# Monotonic Robust Policy Optimization with Model Discrepancy

## Abstract

State-of-the-art deep reinforcement learning (DRL) algorithms tend to overfit in some specific environments due to the lack of data diversity in training. To mitigate the model discrepancy between training and target (testing) environments, domain randomization (DR) can generate plenty of environments with a sufficient diversity by randomly sampling environment parameters in simulator. Though standard DR using a uniform distribution improves the average performance on the whole range of environments, the worst-case environment is usually neglected without any performance guarantee. Since the average and worst-case performance are equally important for the generalization in RL, in this paper, we propose a policy optimization approach for concurrently improving the policy's performance in the average case (i.e., over all possible environments) and the worst-case environment. We theoretically derive a lower bound for the worst-case performance of a given policy over all environments. Guided by this lower bound, we formulate an optimization problem which aims to optimize the policy and sampling distribution together, such that the constrained expected performance of all environments is maximized. We prove that the worst-case performance is monotonically improved by iteratively solving this optimization problem. Based on the proposed lower bound, we develop a practical algorithm, named monotonic robust policy optimization (MRPO), and validate MRPO on several robot control tasks. By modifying the environment parameters in simulation, we obtain environments for the same task but with different transition dynamics for training and testing. We demonstrate that MRPO can improve both the average and worst-case performance in the training environments, and facilitate the learned policy with a better generalization capability in unseen testing environments.

## 1 Introduction

With deep neural network approximation, deep reinforcement learning (DRL) has extended classical reinforcement learning (RL) algorithms to successfully solving complex control tasks, e.g., playing computer games with human-level performance (Mnih et al., 2013; Silver et al., 2018) and continuous robotic control (Schulman et al., 2017). By random exploration, DRL often requires tremendous amounts of data to train a reliable policy. It is thus infeasible for many tasks, such as robotic control and autonomous driving, as training in the real world is not only time-consuming and expensive, but also dangerous. Therefore, training is often conducted on a very limited set of samples, resulting in overfitting and poor generalization capability. One alternative solution is to learn a policy in a simulator (i.e., source/training environment) and then transfer it to the real world (i.e., target/testing environment). Currently, it is impossible to model the exact environment and physics of the real world. For instance, the physical effects like nonrigidity and fluid dynamics are quite difficult to be accurately modeled by simulation. How to mitigate the model discrepancy between the training and target environments remains challenging for the generalization in RL.

To simulate the dynamics of the environment, domain randomization (DR), a simple but effective method is proposed. It randomizes the simulator (e.g., by randomizing the distribution of environment parameters) to generate a variety of environments for training the policy in the source domain. Compared with training in a single environment, recent researches have shown that policies learned through an ensemble of environment dynamics obtained by DR achieve better generalization performance with respect to the expected return. The expected return is referred to as the average per-

formance across all the trajectories sampled from different environments. Since these trajectories, regardless of their performance, are uniformly sampled, the trajectories with the worst performance would severely degrade the overall performance.

In contrast, another line of research on the generalization in RL is from the perspective of control theory, i.e., learning policies that are robust to environment perturbations. Robust RL algorithms learn policies, also using model ensembles produced by perturbing the parameters of the nominal model. EPOpt (Rajeswaran et al., 2017), a representative of them, trains policy solely on the worst performing subset, i.e., trajectories with the worst $\alpha$ percentile of returns, while discarding all the higher performing trajectories. In other words, it seeks a higher worst-case performance at the cost of degradation on the average performance. In general, robust RL algorithms may sacrifice performance on many environment variants and focus only on environments with the worst performance, such that the policy learned will not behave very badly in a previously unseen environment.

In this paper, we focus on the generalization issue in RL, and aim to mitigate the *model discrepancy* of the transition dynamics between the training and target environments. Considering that both the average and worst-case performance are equally important for evaluating the generalization capability of the policy, we propose a policy optimization approach in which the distribution of the sampled trajectories are specifically designed for concurrently improving both the average and worst-case performance. Our main contributions are summarized as follows.

- For a given policy and a wide range of environments, we theoretically derive a lower bound for the worst-case expected return of that policy over all the environments, and prove that maximizing this lower bound (equivalent to maximizing the worst-case performance) can be achieved by solving an average performance maximization problem, subject to constraints that bound the update step in policy optimization and statistical distance between the worst and average case environments. To the best of our knowledge, this theoretical analysis of the relationship between the worst-case and average performance is reported for the first time, which provides a practical guidance for updating policies towards both the worst-case and average performance maximization.

- Trajectories obtained from diverse environments may contribute differently to the generalization capacity of the policy. Therefore, in face of a huge amount of trajectories, the problem that which types of trajectories are likely to mostly affect the generalization performance should be considered. Unlike traditional uniform sampling without the worst-case performance guarantee, and different from the worst $\alpha$ percentile sampling in which the parameter $\alpha$ is empirically preset, we propose a criterion for the sampling trajectory selection based on the proposed worst-case and average performance maximization, with which both the environment diversity and the worst-case environments are taken into account.

- Based on the proposed theorem, we develop a monotonic robust policy optimization (MRPO) algorithm to learn the optimal policy with both the maximum worst-case and average performance. Specifically, MRPO carries out a two-step optimization to update the policy and the distribution of the sampled trajectories, respectively. We further prove that the policy optimization problem can be transformed to trust region policy optimization (TRPO) (Schulman et al., 2015) on all possible environments, such that the policy update can be implemented by the commonly used proximal policy optimization (PPO) algorithm (Schulman et al., 2017). Finally, we prove that by updating the policy with the MRPO, the worst-case expected return can be monotonically increased.

- To greatly reduce the computational complexity, we impose Lipschitz continuity assumptions on the transition dynamics and propose a practical implementation of MRPO. We then conduct experiments on five robot control tasks with variable transition dynamics of environments, and show that MRPO can improve both the average and worst-case performance in the training environments compared to DR and Robust RL baselines, and significantly facilitate the learned policy with a better generalization capability in unseen testing environments.

## 2 BACKGROUND

Under the standard RL setting, the environment is modeled as a Markov decision process (MDP) defined by a tuple $< \mathcal{S}, \mathcal{A}, \mathcal{T}, R >$. $\mathcal{S}$ is the state space and $\mathcal{A}$ is the action space. For the convenience of derivation, we assume they are finite. $\mathcal{T} : \mathcal{S} \times \mathcal{A} \times \mathcal{S} \rightarrow [0, 1]$ is the transition dynamics determined by the environment parameter $p \in \mathcal{P}$, where $\mathcal{P}$ denotes the environment pa-

rameter space. For example in robot control, environment parameter could be physical coefficient that directly affect the control like friction of joints and torso mass. Throughout this paper, by environment $p$, we mean that an environment has the transition dynamics determined by parameter $p$. $R : \mathcal{S} \times \mathcal{A} \rightarrow \mathbb{R}$ is the reward function. At each time step $t$, the agent observes the state $s_t \in \mathcal{S}$ and takes an action $a_t \in \mathcal{A}$ guided by policy $\pi(a_t|s_t)$. Then, the agent will receive a reward $r_t = R(s_t, a_t)$ and the environment shifts from current state $s_t$ to the next state $s_{t+1}$ with probability $\mathcal{T}(s_{t+1}|s_t, a_t, p)$. The goal of RL is to search for a policy $\pi$ that maximizes the expected cumulative discounted reward $\eta(\pi|p) = \mathbb{E}_{\boldsymbol{\tau}}[G(\boldsymbol{\tau}|p)]$, $G(\boldsymbol{\tau}|p) = \sum_{t=0}^{\infty} \gamma^t r_t$. $\boldsymbol{\tau} = \{s_t, a_t, r_t, s_{t+1}\}_{t=0}^{\infty}$ denotes the trajectory generated by policy $\pi$ in environment $p$ and $\gamma \in [0, 1]$ is the discount factor. We can then define the state value function as $V_\pi(s) = \mathbb{E}\left[\sum_{k=0}^{\infty} \gamma^k r_{t+k}|s_t = s\right]$, the action value function as $Q_\pi(s, a) = \mathbb{E}\left[\sum_{k=0}^{\infty} \gamma^k r_{t+k}|s_t = s, a_t = a\right]$, and the advantage function as $A_\pi(s, a) = Q_\pi(s, a) - V_\pi(s)$. We denote the state distribution under environment $p$ and policy $\pi$ as $P_\pi(s|p)$ and that at time step $t$ as $P_\pi^t(s|p)$. During the policy optimization in RL, by updating the current policy $\pi$ to a new policy $\tilde{\pi}$, Schulman et al. (2015) prove that

$$\eta(\tilde{\pi}|p) \geq L_\pi(\tilde{\pi}|p) - \frac{2\lambda\gamma}{(1-\gamma)^2}\beta^2, \quad L_\pi(\tilde{\pi}|p) = \eta(\pi|p) + \mathbb{E}_{s \sim P_\pi(\cdot|p), a \sim \pi(\cdot|s)}\left[\frac{\tilde{\pi}(a|s)}{\pi(a|s)}A_\pi(s, a)\right] \tag{1}$$

where $\lambda = \max_s |\mathbb{E}_{a \sim \pi(a|s)}[A_\pi(s, a)]|$ is the maximum mean advantage following current policy $\pi$ and $\beta = \max_s D_{TV}(\pi(\cdot|s)\|\tilde{\pi}(\cdot|s))$ is the maximum total variation (TV) distance between $\pi$ and $\tilde{\pi}$. The policy's expected return after updating can be monotonically improved by maximizing the lower bound in (1) w.r.t. $\tilde{\pi}$. Based on this and with certain approximation, Schulman et al. (2015) then propose a algorithm named trust region policy optimization (TRPO) that optimizes $\tilde{\pi}$ towards the direction of maximizing $L_\pi(\tilde{\pi})$, subject to the trust region constraint $\beta \leq \delta$.

In standard RL, environment parameter $p$ is fixed without any model discrepancy. While under the domain randomization (DR) settings, because of the existence of model discrepancy, environment parameter should actually be a random variable p following a probability distribution $P$ over $\mathcal{P}$. By introducing DR, the goal of policy optimization is to maximize the mean expected cumulative discounted reward over all possible environment parameters, i.e., $\max_\pi \mathbb{E}_{p \sim P}[\eta(\pi|p)]$.

In face of model discrepancy, our goal is to provide a performance improvement guarantee for the worst-case environments, and meanwhile to improve the average performance over all environments.

**Lemma 1.** *Guided by a certain policy $\pi$, there exists a non-negative constant $C \geq 0$, such that the expected cumulative discounted reward in environment with the worst-case performance satisfies:*

$$\eta(\pi|p_w) - \mathbb{E}_{p \sim P}[\eta(\pi|p)] \geq -C, \tag{2}$$

*where environment $p_w$ corresponds to the worst-case performance, and $C$ is related to $p_w$ and $\pi$.*

*Proof.* See Appendix A.1 for details. $\qquad\square$

**Theorem 1.** *In MDPs where reward function is bounded, for any distribution $P$ over $\mathcal{P}$, by updating the current policy $\pi$ to a new policy $\tilde{\pi}$, the following bound holds:*

$$\eta(\tilde{\pi}|p_w) \geq \mathbb{E}_{p \sim P}[\eta(\tilde{\pi}|p)] - 2|r|_{\max}\frac{\gamma\mathbb{E}_{p \sim P}[\epsilon(p_w\|p)]}{(1-\gamma)^2} - \frac{4|r|_{\max}\alpha}{(1-\gamma)^2}, \tag{3}$$

*where $\epsilon(p_w\|p) \triangleq \max_t \mathbb{E}_{s' \sim P^t(\cdot|p_w)}\mathbb{E}_{a \sim \pi(\cdot|s')}D_{TV}(\mathcal{T}(s|s', a, p_w)\|\mathcal{T}(s|s', a, p))$, environment $p_w$ corresponds to the worst-case performance under the current policy $\pi$, and $\alpha \triangleq \max_t \mathbb{E}_{s' \sim P^t(\cdot|p_w)}D_{TV}(\pi(a|s')\|\tilde{\pi}(a|s'))$.*

*Proof.* See Appendix A.2 for details, and Appendix A.7 for bounded reward function condition. $\square$

In (3), $\epsilon(p_w\|p)$ specifies the model discrepancy between two environments $p_w$ and $p$ in terms of the maximum expected TV distance of their transition dynamics of all time steps in trajectory sampled in environment $p_w$ using policy $\pi$, and $\alpha$ denotes the maximum expected TV distance of two policies along trajectory sampled in environment $p_w$ using policy $\pi$. In general, the RHS of (3) provides a lower-bound for the expected return achieved in the worst-case environment $p_w$, where the first term denotes the mean expected cumulative discounted reward over all environments following the

---

**Algorithm 1** Monotonic Robust Policy Optimization

---

1: Initialize policy $\pi_0$, uniform distribution of environment parameters $U$, number of environment parameters sampled per iteration $M$, maximum number of iterations $N$ and maximum episode length $T$.
2: **for** $k = 0$ to $N - 1$ **do**
3:     Sample a set of environment parameters $\{p_i\}_{i=0}^{M-1}$ according to $U$.
4:     **for** $i = 0$ to $M - 1$ **do**
5:         Sample $L$ trajectories $\{\boldsymbol{\tau}_{i,j}\}_{j=0}^{L-1}$ in environment $p_i$ using $\pi_k$.
6:         Determine $p_w^k = \arg\min_{p_i \in \{p_i\}_{i=0}^{M-1}} \sum_{j=0}^{L-1} G(\boldsymbol{\tau}_{i,j}|p_i)/L$.
7:         Compute $\hat{\mathcal{E}}(p_i, \pi_k) = \frac{\sum_{j=0}^{L-1} G(\boldsymbol{\tau}_{i,j}|p_i)}{L} - \frac{2|r|_{max}\gamma\epsilon(p_i\|p_w^k)}{(1-\gamma)^2}$ for environment $p_i$.
8:     **end for**
9:     Select the trajectory set $\mathbb{T} = \{\boldsymbol{\tau}_i : \hat{\mathcal{E}}(p_i, \pi_k) \geq \hat{\mathcal{E}}(p_w^k, \pi_k)\}$.
10:     Use PPO for policy optimization on $\mathbb{T}$ to get the updated policy $\pi_{k+1}$.
11: **end for**

---

sampling distribution $P$, while the other two terms can be considered as penalization on a large TV distance between the worst-case environment $p_w$ and the average case, and a large update step from the current policy $\pi$ to the new policy $\tilde{\pi}$, respectively. Therefore, by maximizing this lower bound, we can improve the worst-case performance, which in practice is equivalent to the following constrained optimization problem with two constraints:

$$\max_{\tilde{\pi}, P} \mathbb{E}_{\mathrm{p}\sim P}\left[\eta(\tilde{\pi}|p)\right] \quad \text{s.t.} \quad \alpha \leq \delta_1, \quad \mathbb{E}_{\mathrm{p}\sim P}\left[\epsilon(p_w\|p)\right] \leq \delta_2. \tag{4}$$

The optimization objective is to maximize the mean expected cumulative discounted reward over all possible environments, by updating not only the policy $\tilde{\pi}$, but the environment parameter's sampling distribution $P$. The first constraint imposes a similar trust region to TRPO (Schulman et al., 2017) that constrains the update step in policy optimization. In addition, we further propose a new trust region constraint on the sampling distribution $P$ that the TV distance between the worst-case environment $p_w$ and average case over $P$ is bounded, such that by achieving the optimization objective in (4), the worst-case performance is also improved.

To solve the constrained optimization problem in (4), we need to seek for the optimal policy $\tilde{\pi}$ and the distribution $P$ of the sampled trajectories. In practice, we carry out a two-step optimization procedure to simplify the computational complexity. First, we fix the policy by letting $\tilde{\pi} = \pi$, and optimize the objective in (4) w.r.t. the distribution $P$. In this case, we no longer need to consider the first constraint on the policy update, and thus can convert the second constraint on the sampling distribution into the objective with the guidance of Theorem 1, formulating the following unconstrained optimization problem:

$$\max_{P} \mathbb{E}_{\mathrm{p}\sim P}\left[\mathcal{E}(p, \tilde{\pi})\right], \tag{5}$$

where we denote $\mathcal{E}(p, \tilde{\pi}) \triangleq \eta(\tilde{\pi}|p) - \frac{2|r|_{max}\gamma\epsilon(p\|p_w)}{(1-\gamma)^2}$. The first term in $\mathcal{E}(p, \tilde{\pi})$ indicates policy $\tilde{\pi}$'s performance in environment $p$, while the second term measures the model discrepancy between environment $p$ and $p_w$. Since the objective function in (5) is linear to $P$, we can update $P$ by assigning a higher probability to environment $p$ with higher $\mathcal{E}(p, \tilde{\pi})$. As a consequence, sampling according to $\mathcal{E}(p, \tilde{\pi})$ would increase the sampling probability of environments with both poor and good-enough performance, and avoid being trapped in the worst-case environment. Specifically, we propose to select samples from environment $p$ that meets $\mathcal{E}(p, \tilde{\pi}) \geq \mathcal{E}(p_w, \tilde{\pi})$ for the training of policy $\tilde{\pi}$, which is equivalent to assigning a zero probability to the other samples.

In the second step, we target at optimizing the policy $\pi$ with the updated distribution $P$ fixed, i.e., the following optimization problem:

$$\max_{\tilde{\pi}} \mathbb{E}_{\mathrm{p}\sim P}\left[\eta(\tilde{\pi}|p)\right] \quad \text{s.t.} \quad \alpha \leq \delta_1 \tag{6}$$

Optimization in (6) can be transformed to a trust region robust policy optimization similar to TRPO and solve it practically with PPO (refer to Appendix A.3 and Schulman et al. (2017) for more information). To summarize, we propose a monotonic robust policy optimization (MRPO) in Algorithm1.

At each iteration $k$, we uniformly sample $M$ environments and run a trajectory for each sampled environment. For each environment $p_i$, we sample $L$ trajectories $\{\boldsymbol{\tau}_{i,j}\}_{j=1}^{L}$, approximate $\eta(\pi_k|p_i)$ with $\sum_{j=0}^{L-1} G(\boldsymbol{\tau}_{i,j}|p_i)/L$, and determine the worst-case environment $p_w$ based on $\sum_{j=0}^{L-1} G(\boldsymbol{\tau}_{i,j}|p_i)/L$ of a given set of environments $p_i{}_{i=0}^{M-1}$, We then optimize the policy with PPO on the selected trajectory subset $\mathbb{T}$ according to $\mathcal{E}(p_i, \pi_k)$ and $\mathcal{E}(p_w^k, \pi_k)$.

We now formally show that by maximizing the lower bound provided in Theorem1, the worst-case performance within all the environments can be monotonically improved by MRPO.

**Theorem 2.** *The sequence of policy $\{\pi_1, \pi_2, \ldots, \pi_N\}$ generated by Algorithm1 is guaranteed with the monotonic worst-case performance improvement, i.e.,*

$$\eta(\pi_1|p_w^1) \leq \eta(\pi_2|p_w^2) \leq \cdots \leq \eta(\pi_N|p_w^N), \tag{7}$$

*where $p_w^k$ denotes the parameter of environment with the worst-case performance guided by the current policy $\pi_k$ at iteration $k$.*

*Proof.* See Appendix A.4 for details. □

## 3 PRACTICAL IMPLEMENTATION USING SIMULATOR

Motivated by Theorem 1, we propose Algorithm 1 that provably promotes monotonic improvement for the policy's performance in the worst-case environment according to Theorem 2. However, Theorem 1 imposes calculation of $\epsilon(p_w\|p)$ that requires the estimation of the expected total variation distance between the worst-case environment and every other sampled environment at each time step. Estimation by sampling takes exponential complexity. Besides, in the model-free setting, we unfortunately have no access to analytical equation of the environment's transition dynamics. Hence, computation for the total variation distance between two environments is unavailable. Under the deterministic state transition, an environment will shift its state with a probability of one. Hence, we can set state of one environment's simulator to the state along the trajectory from the other environment, and take the same action. We then can compare the next state and compute the total variation distance step by step. Though feasible, this method requires large computational consumption. In this section, we propose instead a practical implementation for Algorithm 1.

According to Appendix A.9, we first make a strong assumption that the transition dynamics model is $L_p$-Lipschitz in terms of the environment parameter $p$:

$$\|\mathcal{T}(s|s', a, p) - \mathcal{T}(s|s', a, p_w)\| \leq L_p\|p - p_w\|. \tag{8}$$

Then, we can simplify the calculation of $\epsilon(p_w\|p)$ via:

$$\epsilon(p_w\|p) \leq \max_t \mathbb{E}_{s'\sim P^t(\cdot|p_w, \pi)} \mathbb{E}_{a\sim\pi(\cdot|s')} \frac{1}{2} \sum_s L_p\|p - p_w\| = \frac{1}{2} \sum_s L_p\|p - p_w\|. \tag{9}$$

It can be seen from the expression of $\epsilon(p_w\|p)$ that it measures the transition dynamics distance between the worst-case environment $p_w$ and a specific environment $p$. In simulator, the environment's transition dynamics would vary with the environment parameters, such as the friction and mass in robot simulation. Hence the difference between environment parameters can reflect the distance between the transition dynamics. In addition, if we use in practice the penalty coefficient of $\epsilon(p_w\|p)$ as recommended by Theorem 1, the subset $\mathbb{T}$ would be very small. Therefore, we integrate it with $L_p$ as a tunable hyperparameter $\kappa$, and propose a practical version of MRPO in Algorithm 2 in Appendix A.6.

## 4 EXPERIMENTS

We now evaluate the proposed MRPO in five robot control benchmarks designed for evaluation of generalization under changeable dynamics. These five environments are modified by the open-source generalization benchmarks (Packer et al., 2018) based on the complex robot control tasks introduced in (Schulman et al., 2017). We compare MRPO with two baselines, PPO-DR and PW-DR, respectively. In PPO-DR, PPO is applied for the policy optimization in DR. In PW-DR, we use

Table 1: Range of parameters for each environment.

| Environment | Parameters | Training Range | Testing Range |
|---|---|---|---|
| Walker2d/Hopper/HalfCheetah | Density | $[750, 1250]$ | $[1, 750]$ |
| | Friction | $[0.5, 1.1]$ | $[0.2, 0.5]$ |
| HalfcheetahBroadRange | Density | $[750, 1250]$ | $[1, 750]$ |
| | Friction | $[0.2, 2.25]$ | $[0.05, 0.2]$ |
| InvertedDoublePendulum | Pole1 length | $[0.50, 1.00]$ | $[1.00, 1.10]$ |
| | Pole2 length | $[0.50, 1.00]$ | $[1.00, 1.10]$ |
| Cartpole | Force magnitude | $[1.00, 20.00]$ | $10$ |
| | Pole length | $[0.05, 1.00]$ | $[1.00, 6.00]$ |
| | Pole mass | $[0.01, 1.00]$ | $[1.00, 6.00]$ |

purely trajectories from the $10\%$ worst-case environments for training and still apply PPO for the policy optimization. Note that PW-DR is the implementation of EPOpt algorithm proposed in (Rajeswaran et al., 2017) without the value function baseline. Since this value function baseline in EPOpt is not proposed for the generalization improvement (in fact it could incur a performance improvement for all the policy optimization algorithms), we do not adopt it in PW-DR. We utilize two 64-unit hidden layers to construct the policy network and value function in PPO. For MRPO, we use the practical implementation as described in Algorithm 2. Further note that during the experiments, we find that environments generating poor performance would be far away from each other in terms of the TV distance between their transition dynamics with the dimension of changeable parameters increasing. In other words, a single worst-case environment usually may not represent all the environments where the current policy performs very poorly. Hence, we choose the $10\%$ worst-case environments to replace the calculation of $\hat{\mathcal{E}}'(p_w^k, \pi_k)$ in Algorithm 2. That is, the trajectories we add to the subset $\mathbb{T}$ should have the $\hat{\mathcal{E}}'(p_i, \pi_k)$ greater than and equal to those of all the $10\%$ worst-case environments. From the expression of $\hat{\mathcal{E}}'(p_i, \pi_k) = \sum_{j=0}^{L-1} G(\boldsymbol{\tau}_{i,j}|p_i)/L - \kappa\|p_i - p_w^k\|$ for environment $p_i$, it can be seen that the trajectory selection is based on a trade-off between the performance and the distance to the worst-case environment, as we described in detail in the paragraph under (5). Our experiments are designed to investigate the following two classes of questions:

- Can MRPO effectively improve the policy's worst-case and average performance over the whole range of environments during training. Compared to DR, will the policy's average performance degrade by using MRPO? Will MRPO outperform PW-DR in the worst-case environments?

- How does the performance of trained robust policy using MRPO degrade when employed in environments with unseen dynamics? And what determines the generalization performance to unseen environments during training?

## 4.1 TRAINING PERFORMANCE WITH DIFFERENT DYNAMICS

The five robot control tasks are as follows. (1) **Walker2d**: control a 2D bipedal robot to walk; (2) **Hopper**: control a 2D one-legged robot to hop as fast as possible; (3) **HalfCheetah**: control a 2D cheetah robot to run (Brockman et al., 2016); (4) **InvertedDoublePendulum**: control a cart (attached to a two-link pendulum system by a joint) by applying a force to prevent the two-link pendulum from falling over; and (5) **Cartpole**: control a cart (attached to a pole by a joint) by applying a force to prevent the pole from falling over. In robot control, the environment dynamics are directly related to the value of some physical coefficients. For example, if sliding friction of the joints is large, it will be more difficult for the agent to manipulate the robot compared to a smaller sliding friction. Hence, a policy that performs well for a small friction may not by generalized to the environment with a large friction due to the change of dynamics. In the simulator, by randomizing certain environment parameters, we can obtain a set of environments with the same goal but different dynamics (Packer et al., 2018). The range of parameters that we preset for training of each environment is shown in Table 1.

We run the training process for the same number of iterations $N$ sampled from preset range of environment parameters. At each iteration $k$, we generate trajectories from $M = 100$ environments sampled according to a uniform distribution $U$. The results are obtained by running each algorithm

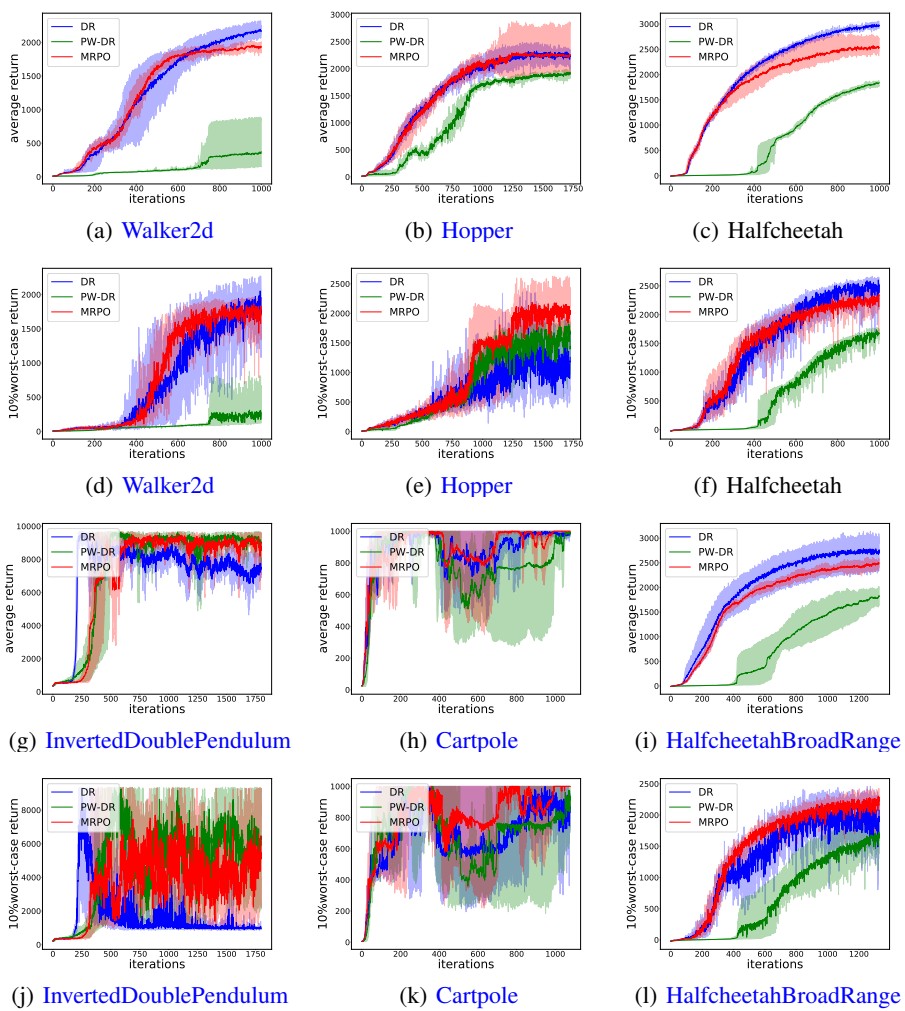

Figure 1: Training curves of average return and $10\%$ worst-case return.

Table 2: Average and worst-case performance in training, where W, H, C, CP, I and CB refer to Walker2d, Hopper, Halfcheetah, Cartpole, InvertedDoublePendulum and HalfcheetahBroadRange.

| Algorithm | Average W | Worst W | Average H | Worst H | Average C | Worst C |
|---|---|---|---|---|---|---|
| MRPO | $1930.0 \pm 66.7$ | $1702.5 \pm 185.9$ | $\mathbf{2257.4 \pm 423.5}$ | $\mathbf{2048.0 \pm 246.8}$ | $2537.0 \pm 98.3$ | $2340.0 \pm 61.2$ |
| DR | $\mathbf{2170.0 \pm 57.9}$ | $\mathbf{1772.5 \pm 162.2}$ | $2250.0 \pm 80.4$ | $1254.0 \pm 348.5$ | $\mathbf{2970.0 \pm 30.0}$ | $\mathbf{2512.5 \pm 120.9}$ |
| PW-DR | $372.0 \pm 291.2$ | $273.0 \pm 230.0$ | $1903.3 \pm 49.2$ | $1440.0 \pm 243.9$ | $1810.0 \pm 41.2$ | $1677.5 \pm 72.3$ |

| Algorithm | Average I | Worst I | Average CP | Worst CP | Average CB | Worst CB |
|---|---|---|---|---|---|---|
| MRPO | $9185.2 \pm 165.6$ | $5137.0 \pm 1421.1$ | $\mathbf{998.7 \pm 1.7}$ | $\mathbf{988.4 \pm 15.4}$ | $2363.3 \pm 82.6$ | $\mathbf{2166.7 \pm 70.4}$ |
| DR | $8252.6 \pm 439.8$ | $1354.5 \pm 381.2$ | $995.5 \pm 5.6$ | $961.6 \pm 48.3$ | $\mathbf{2646.7 \pm 303.5}$ | $1883.3 \pm 103.7$ |
| PW-DR | $\mathbf{9222.2 \pm 173.3}$ | $\mathbf{5450.4 \pm 1687.3}$ | $895.4 \pm 147.2$ | $791.4 \pm 288.9$ | $1456.7 \pm 301.3$ | $1278.7 \pm 215.9$ |

with five different random seeds. The average return is computed over the returns of $M$ sampled environments at each iteration $k$. We show the training curves of Walker2d, Hopper and Halfcheetah, InvertedDoublePendulum, Cartpole, and HalfcheetahBroadRange, respectively, in Figs. 1(a)-1(c) and 1(g)-1(i). In Figure 1, the solid curve is used to represent the average performance of each algorithm on all the five seeds, while the shaded-area denotes the standard error. It is seen that DR can steadily improve the average performance on the whole training range as expected, while MRPO does not significantly degrade the average performance in all the the tasks. PW-DR, on the other hand, focuses on the worst-case performance optimization, leading to an obvious degradation of av-

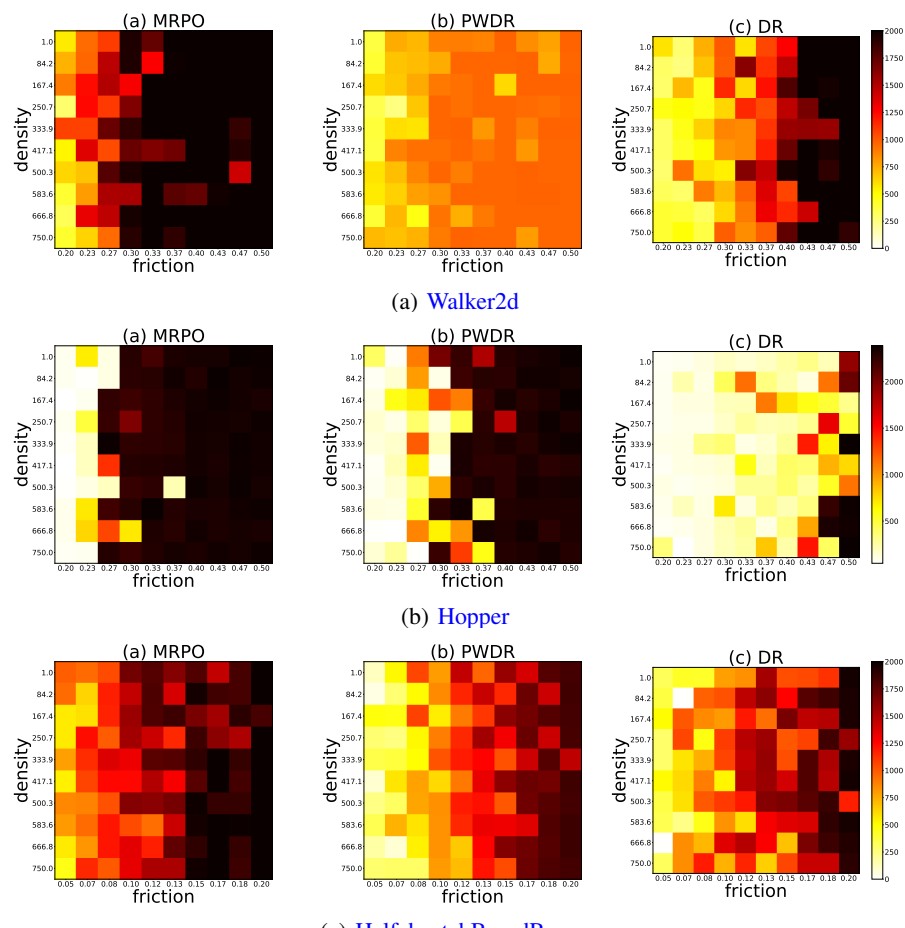

Figure 2: Heatmap of return in unseen environments on Waler2d, Hopper and Halfcheetah with policies trained by MRPO, PW-DR and DR in the training environments.

erage performance on Hopper and Halfcheetah and failure on Walker2d. We measure the worst-case performance by computing the worst $10\%$ performance in all the sampled environments at iteration $k$ and the corresponding training curves are illustrated in Figs. 1(d)-1(f) and 1(j)-1(l), respectively. It can be observed that MRPO presents the best worst-case performance on Hopper, Cartpole and HalfcheetahBroadRange, while DR neglects the optimization on its worst-case performance and performs badly in these three tasks. PW-DR shows limited improvement of the worst-case performance compared to MRPO on Hopper, Halfcheetah, Cartpole and HalfcheetahBroadRange, and failure on Walker2d. Comparing Figs. 1(f) and 1(l), it can be concluded that the original parameter range we set for the Halfcheetah task (e.g., the friction range of $[0.5, 1.1]$) was too narrow to cause seriously poor performance on the $10\%$ worst-case environments. By enlarging the friction range from $[0.5, 1.1]$ to $[0.2, 2.5]$ for HalfcheetahBroadRange, the training curves in Fig. 1(l) can clearly demonstrate that MRPO outperforms the other baselines. The tabular comparison of the average and worst-case performance achieved during training by different algorithms in different tasks can be found in Table 2.

## 4.2 GENERALIZATION TO UNSEEN ENVIRONMENTS

MRPO has been demonstrated in theoretical analysis to optimize both average and worst-case performance during training. Here, we carry out experiments to show that MRPO can generalize to a broader range of unseen environments in testing. To this end, we compare the testing performance on some unseen environments of Walker2d, Hopper and HalfcheetahBroadRange with the best policies obtained by MRPO, DR and PW-DR from training, with the range of parameters set for testing showin in Table 1. It is observed that policies all degrade with the decrease of friction, while the impact of unseen density is not that obvious as the friction. The heatmap of return achieved in all the testing Hopper environments by each algorithm is shown and compared in Fig. 2. It can be seen that

MRPO has better generalization ability to the unseen environments, while DR can hardly generalize in testing. Compared to PW-DR, MRPO has a broader generalization range, from which We remark that both the worst-case and average performance during training are crucial for the generalization to an unseen environment.

## 5 RELATED WORK

With the success of RL in recent years, plenty of works have focused on how to improve the generalization ability for RL. Learning a policy that is robust to the worst-case environment is one strategy. Based on theory of $\mathcal{H}_\infty$ control (Zhou et al., 1996), robust RL takes into account the disturbance of environment parameters and model it as an adversary that is able to disturb transition dynamics in order to prevent the agent from achieving higher rewards (Morimoto & Doya, 2005). The policy optimization is then formulated as a zero-sum game between the adversary and the RL agent. Pinto et al. (2017) incorporate robust RL to DRL method, which improves robustness of DRL in complex robot control tasks. To solve robust RL problem, robust dynamic programming formulates a robust value function and proposes accordingly a robust Bellman operator (Iyengar, 2005; Mankowitz et al., 2020). The optimal robust policy can then be achieved by iteratively applying the robust Bellman operator in a similar way to the standard value iteration (Sutton & Barto, 2018). Besides, Rajeswaran et al. (2017) leverage data from the worst-case environments as adversarial samples to train a robust policy. However, the aforementioned robust formulations will lead to an unstable learning. What's worse, the overall improvement of the average performance over the whole range of environments will also be stumbled by their focus on the worst-case environments. In contrast, in addition to the worst-case formulation, we also aim to improve the average performance.

For generalization across different state spaces, an effective way is domain adaptation, which maps different state space to a common embedding space. The policy trained on this common space can then be easily generalized to a specific environment (Higgins et al., 2017b; James et al., 2019; Ammar et al., 2015) through a learned mapping, with certain mapping methods, such as $\beta$-VAE (Higgins et al., 2017a), cGAN (Isola et al., 2017), and manifold alignment (Wang & Mahadevan, 2009).

Function approximation enables RL to solve complex tasks with high-dimensional state and action spaces, which also incurs inherent generalization issue under supervised learning. Deep neural network (DNN) suffers overfitting due to the distribution discrepancy between training and testing sets. $l_2$-regularization, dropout and dataset augmentation (Goodfellow et al., 2016) play an significant role for generalization in deep learning, which have also enabled improvement of policy's generalization on some specifically designed environments (Cobbe et al., 2019; Farebrother et al., 2018).

In terms of the theoretical analysis, Murphy (2005) provide a generalization error bound for $Q$-learning, where by generalization error they mean the distance between expected discounted reward achieved by converged $Q$-learning policy and the optimal policy. Wang et al. (2019) analyze the generalization gap in reparameterizable RL limited to the Lipschitz assumptions on transition dynamics, policy and reward function. For monotonic policy optimization in RL, Schulman et al. (2015) propose to optimize a constrained surrogate objective, which can guarantee the performance improvement of updated policy. In the context of model-based RL, Janner et al. (2019); Luo et al. (2019) formulate the lower bound for a certain policy's performance on true environment in terms of the performance on the learned model. It can therefore monotonically improve the performance on true environment by maximizing this lower bound. Different from this, the proposed MRPO in this work can guarantee the robustness of the policy in terms of the monotonically increased worst-case performance, and also improve the average performance.

## 6 CONCLUSION

In this paper, we have proposed a robust policy optimization approach, named MRPO, for improving both the average and worst-case performance of policies. Specifically, we theoretically derived a lower bound for the worst-case performance of a given policy over all environments, and formulated an optimization problem to optimize the policy and sampling distribution together, subject to constraints that bounded the update step in policy optimization and statistical distance between the worst and average case environments. We proved that the worst-case performance was monotonically improved by iteratively solving this optimization problem. We have validated MRPO on several robot control tasks, demonstrating a performance improvement on both the worst and average case environments, as well as a better generalization ability to a wide range of unseen environments.

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

## A  APPENDIX

### A.1  PROOF OF LEMMA 1

*Proof.* First, we define $\eta(\pi|p_w) - \max_{p\in\mathcal{P}} \eta(\pi|p) \triangleq -C_0$, where $C_0 \geq 0$ depends on $\pi$ and $\mathcal{P}$. Then, given a policy $\pi$ and any environment $p \in \mathcal{P}$, and for any non-negative constant $C \geq C_0 \geq 0$, we thus have:

$$\eta(\pi|p_w) - \mathbb{E}_{\mathrm{p}\sim P}\left[\eta(\pi|p)\right] \geq \eta(\pi|p_w) - \max_{p\in\mathcal{P}}\eta(\pi|p) = -C_0 \geq -C. \tag{10}$$

$\square$

### A.2  PROOF OF THEOREM 1

**Lemma 2.** *For any two joint distribution $P_1(\mathrm{x},\mathrm{y}) = P_1(\mathrm{x})P_1(\mathrm{y}|\mathrm{x})$ and $P_2(\mathrm{x},\mathrm{y}) = P_2(\mathrm{x})P_2(\mathrm{y}|\mathrm{x})$ over $\mathrm{x}$ and $\mathrm{y}$, we can bound the total variation distance of them by:*

$$D_{TV}(P_1(\mathrm{x},\mathrm{y})\|P_2(\mathrm{x},\mathrm{y})) \leq D_{TV}(P_1(\mathrm{x})|P_2(\mathrm{x})) + \max_x D_{TV}(P_1(\mathrm{y}|x)\|P_2(\mathrm{y}|x)) \tag{11}$$

*Proof.*

$$D_{TV}(P_1(\mathrm{x},\mathrm{y})\|P_2(\mathrm{x},\mathrm{y})) = \frac{1}{2}\sum_{x,y}|P_1(x,y) - P_2(x,y)| \tag{12}$$

$$= \frac{1}{2}\sum_{x,y}|P_1(x)P_1(y|x) - P_2(x)P_2(y|x)| \tag{13}$$

$$= \frac{1}{2}\sum_{x,y}|P_1(x)P_1(y|x) - P_1(x)P_2(y|x) + P_1(x)P_2(y|x) - P_2(x)P_2(y|x)| \tag{14}$$

$$\leq \frac{1}{2}\sum_{x,y}P_1(x)|P_1(y|x) - P_2(y|x)| + \frac{1}{2}\sum_x|P_1(x) - P_2(x)| \tag{15}$$

$$= \mathbb{E}_{x\sim P_1}D_{TV}(P_1(\mathrm{y}|x)\|P_2(\mathrm{y}|x)) + D_{TV}(P_1(\mathrm{x})|P_2(\mathrm{x})) \tag{16}$$

$\square$

**Lemma 3.** *Suppose the initial state distributions $P_1^0(\mathrm{s})$ and $P_2^0(\mathrm{s})$ are the same. Then the distance in the state marginal at time step $t$ is bounded as:*

$$D_{TV}(P_1^t(\mathrm{s})\|P_2^t(\mathrm{s})) \leq t\max_t \mathbb{E}_{s'\sim P_1^t}D_{TV}(P_1(\mathrm{s}|s')\|P_2(\mathrm{s}|s')) \tag{17}$$

*Proof.*

$$|P_1^t(s) - P_2^t(s)| = |\sum_{s'}P_1(\mathrm{s}_t = s|s')P_1^{t-1}(s') - \sum_{s'}P_2(\mathrm{s}_t = s|s')P_2^{t-1}(s')| \tag{18}$$

$$\leq \sum_{s'}|P_1(\mathrm{s}_t = s|s')P_1^{t-1}(s') - P_2(\mathrm{s}_t = s|s')P_2^{t-1}(s')| \tag{19}$$

$$= \sum_{s'}|P_1(\mathrm{s}_t = s|s')P_1^{t-1}(s') - P_2(\mathrm{s}_t = s|s')P_1^{t-1}(s') \tag{20}$$

$$+ P_2(\mathrm{s}_t = s|s')P_1^{t-1}(s') - P_2(\mathrm{s}_t = s|s')P_2^{t-1}(s')| \tag{21}$$

$$\leq \mathbb{E}_{s'\sim P_1^{t-1}}|P_1(s|s') - P_2(s|s')| + \sum_{s'}P_2(s|s')|P_1^{t-1}(s') - P_2^{t-1}(s')| \tag{22}$$

$$D_{TV}(P_1^t(\mathbf{s})\|P_2^t(\mathbf{s})) \leq \frac{1}{2}\sum_s |P_1^t(s) - P_2^t(s)| \tag{23}$$

$$\leq \frac{1}{2}\sum_s \left(\mathbb{E}_{s'\sim P_1^{t-1}}|P_1(s|s') - P_2(s|s')| + \sum_{s'} P_2(s|s')|P_1^{t-1}(s') - P_2^{t-1}(s')|\right) \tag{24}$$

$$= \mathbb{E}_{s'\sim P_1^{t-1}} D_{TV}(P_1(\mathbf{s}|s')\|P_2(\mathbf{s}|s')) + D_{TV}(P_1^{t-1}(\mathbf{s}')\|P_2^{t-1}(\mathbf{s}')) \tag{25}$$

$$\leq \sum_{i=1}^t \mathbb{E}_{s'\sim P_1^{i-1}} D_{TV}(P_1(\mathbf{s}|s')\|P_2(\mathbf{s}|s')) \tag{26}$$

$$\leq t \max_t \mathbb{E}_{s'\sim P_1^t} D_{TV}(P_1(\mathbf{s}|s')\|P_2(\mathbf{s}|s')) \tag{27}$$

$\square$

**Theorem 3.** *(A modified version of Theorem 1.) For any distribution $P$ over $\mathcal{P}$, by updating the current policy $\pi$ to a new policy $\tilde{\pi}$, the following bound holds:*

$$\eta(\tilde{\pi}|p_w) - \mathbb{E}_{\mathbf{p}\sim P}\left[\eta(\tilde{\pi}|p)\right] \geq -2|r|_{\max}\frac{\gamma\mathbb{E}_{\mathbf{p}\sim P}\left[\epsilon(p_w\|p)\right]}{(1-\gamma)^2} - \frac{4|r|_{\max}\alpha}{(1-\gamma)^2}, \tag{28}$$

*where $\epsilon(p_w\|p) \triangleq \max_t \mathbb{E}_{s'\sim P^t(\cdot|p_w)}\mathbb{E}_{\mathbf{a}\sim\pi(\cdot|s')} D_{TV}(\mathcal{T}(\mathbf{s}|s',\mathbf{a},p_w)\|\mathcal{T}(\mathbf{s}|s',\mathbf{a},p))$, environment $p_w$ corresponds to the worst-case performance, and $\alpha \triangleq \max_t \mathbb{E}_{s'\sim P^t(\cdot|p_w)} D_{TV}(\pi(\mathbf{a}|s')\|\tilde{\pi}(\mathbf{a}|s'))$.*

*Proof.* We can rewrite the LHS of (2) as:

$$\eta(\tilde{\pi}|p_w) - \mathbb{E}_{\mathbf{p}\sim P}\left[\eta(\tilde{\pi}|p)\right] = \eta(\tilde{\pi}|p_w) - \eta(\pi|p_w) + \eta(\pi|p_w) - \mathbb{E}_{\mathbf{p}\sim P}\left[\eta(\tilde{\pi}|p)\right].$$

For the last two term, we have

$$\mathbb{E}_{\mathbf{p}\sim P}|\eta(\pi|p_w) - \eta(\tilde{\pi}|p)| = \mathbb{E}_{\mathbf{p}\sim P}|\sum_t \gamma^t \sum_{s,a}(P^t(s,a|p_w) - P^t(s,a|p))R(s,a))|$$

$$\leq \mathbb{E}_{\mathbf{p}\sim P}\sum_t \gamma^t \sum_{s,a}|P^t(s,a|p_w) - P^t(s,a|p)||R(s,a)|$$

$$= 2|r|_{\max}\sum_t \gamma^t \mathbb{E}_{\mathbf{p}\sim P}\left[D_{TV}(P^t(\mathbf{s},\mathbf{a}|p_w)\|P^t(\mathbf{s},\mathbf{a}|p))\right], \tag{29}$$

where $P^t(\mathbf{s},\mathbf{a}|p_w) = \pi(\mathbf{a}|s)P^t(\mathbf{s}|p_w)$ and $P^t(\mathbf{s},\mathbf{a}|p) = \tilde{\pi}(\mathbf{a}|s)P^t(\mathbf{s}|p)$. By Lemma 2, we have

$$\mathbb{E}_{\mathbf{p}\sim P}\left[D_{TV}(P^t(\mathbf{s},\mathbf{a}|p_w)|P^t(\mathbf{s},\mathbf{a}|p))\right]$$

$$\leq \mathbb{E}_{\mathbf{s}\sim P^t(\cdot|p_w)} D_{TV}(\pi(\mathbf{a}|\mathbf{s})\|\tilde{\pi}(\mathbf{a}|\mathbf{s})) + \mathbb{E}_{\mathbf{p}\sim P}\left[D_{TV}(P^t(\mathbf{s}|p_w)\|P^t(\mathbf{s}|p))\right]. \tag{30}$$

Note that

$$P(\mathbf{s}|s',p_w) = \sum_a \mathcal{T}(\mathbf{s}|s',a,p_w)\pi(a|s') \tag{31}$$

$$P(\mathbf{s}|s',\mathbf{p}) = \sum_a \mathcal{T}(\mathbf{s}|s',a,\mathbf{p})\tilde{\pi}(a|s'). \tag{32}$$

Similar to Lemma 2, we have

$$D_{TV}(P(\mathbf{s}|s',p_w)\|P(\mathbf{s}|s',\mathbf{p})) \tag{33}$$

$$= \frac{1}{2}\sum_s\sum_a |\mathcal{T}(s|s',a,p_w)\pi(a|s') - \mathcal{T}(s|s',a,\mathbf{p})\tilde{\pi}(a|s')| \tag{34}$$

$$\leq \frac{1}{2}\sum_s\sum_a |\mathcal{T}(s|s',a,p_w) - \mathcal{T}(s|s',a,\mathbf{p})|\pi(a|s') + \frac{1}{2}\sum_s\sum_a \mathcal{T}(s|s',a,\mathbf{p})|\pi(a|s') - \tilde{\pi}(a|s')| \tag{35}$$

$$= \mathbb{E}_{\mathbf{a}\sim\pi(\cdot|s')} D_{TV}(\mathcal{T}(\mathbf{s}|s',a,p_w)\|\mathcal{T}(\mathbf{s}|s',a,\mathbf{p})) + D_{TV}(\pi(\mathbf{a}|s')\|\tilde{\pi}(\mathbf{a}|s')) \tag{36}$$

By Lemma 3, we have

$$\mathbb{E}_{p \sim P} \left[ D_{TV}(P^t(s|p_w) \| P^t(s|p)) \right]$$

$$\leq t\mathbb{E}_{p \sim P} \max_t \mathbb{E}_{s' \sim P^t(\cdot|p_w)} D_{TV}(P(s|s', p_w) \| P(s|s', p))$$

$$\leq t\mathbb{E}_{p \sim P} \max_t \mathbb{E}_{s' \sim P^t(\cdot|p_w)} \mathbb{E}_{a \sim \pi(\cdot|s')} D_{TV}(\mathcal{T}(s|s', a, p_w) \| \mathcal{T}(s|s', a, p))$$

$$+ t \max_t \mathbb{E}_{s' \sim P^t(\cdot|p_w)} D_{TV}(\pi(a|s') \| \tilde{\pi}(a|s')) \tag{37}$$

Since $\epsilon(p\|p_w) = \max_t \mathbb{E}_{s' \sim P^t(\cdot|p_w)} \mathbb{E}_{a \sim \pi(\cdot|s')} D_{TV}(\mathcal{T}(s|s', a, p_w) \| \mathcal{T}(s|s', a, p))$ and $\alpha = \max_t \mathbb{E}_{s' \sim P^t(\cdot|p_w)} D_{TV}(\pi(a|s') \| \tilde{\pi}(a|s'))$, combining (29), (30) and (37), and referring to Jensen's inequality, we have

$$|\eta(\pi|p_w) - \mathbb{E}_{p \sim P}\eta(\tilde{\pi}|p)|$$

$$\leq \mathbb{E}_{p \sim P} |\eta(\pi|p_w) - \eta(\tilde{\pi}|p)|$$

$$\leq 2|r|_{\max} \sum_t \gamma^t \mathbb{E}_{p \sim P} \left[ (t+1) \max_t \mathbb{E}_{s' \sim P^t(\cdot|p_w)} D_{TV}(\pi(a|s') \| \tilde{\pi}(a|s')) \right.$$

$$\left. + t \max_t \mathbb{E}_{s' \sim P^t(\cdot|p_w)} \mathbb{E}_{a \sim \pi(\cdot|s')} D_{TV}(\mathcal{T}(s|s', a, p_w) \| \mathcal{T}(s|s', a, p)) \right]$$

$$= 2|r|_{\max} \sum_t \gamma^t \left[ t(\mathbb{E}_{p \sim P} \left[ \epsilon(p_w\|p) \right] + \alpha) + \alpha \right]$$

$$= 2|r|_{\max} \left[ \frac{\gamma \mathbb{E}_{p \sim P} \left[ \epsilon(p_w\|p) \right]}{(1-\gamma)^2} + \frac{\alpha}{(1-\gamma)^2} \right]. \tag{38}$$

With policy $\pi$ be updated to $\tilde{\pi}$, $\eta(\pi|p_w) \leq \mathbb{E}_{p \sim P} \left[ \eta(\tilde{\pi}|p) \right]$. Then,

$$\eta(\pi|p_w) - \mathbb{E}_{p \sim P} \left[ \eta(\tilde{\pi}|p) \right] \geq -2|r|_{\max} \left[ \frac{\gamma \mathbb{E}_{p \sim P} \left[ \epsilon(p_w\|p) \right]}{(1-\gamma)^2} + \frac{\alpha}{(1-\gamma)^2} \right]. \tag{39}$$

Similar to the derivation of (38) and refer to Janner et al. (2019), we have

$$\eta(\tilde{\pi}|p_w) - \eta(\pi|p_w) \geq -\frac{2|r|_{\max}\alpha}{(1-\gamma)^2}. \tag{40}$$

Combining the above results, we end up with the proof

$$\eta(\tilde{\pi}|p_w) - \mathbb{E}_{p \sim P} \left[ \eta(\tilde{\pi}|p) \right] \geq -2|r|_{\max} \frac{\gamma \mathbb{E}_{p \sim P} \left[ \epsilon(p_w\|p) \right]}{(1-\gamma)^2} - \frac{4|r|_{\max}\alpha}{(1-\gamma)^2}. \tag{41}$$

$$\square$$

### A.3 DERIVATION OF POLICY OPTIMIZATION STEP

In the policy optimization step, we aim to solve the following optimization problem:

$$\max_{\tilde{\pi}} \mathbb{E}_{p \sim P} \left[ \eta(\tilde{\pi}|p) \right] \quad \text{s.t.} \quad \alpha \leq \delta_1. \tag{42}$$

Referring to (1), we have:

$$\mathbb{E}_{p \sim P} \left[ \eta(\tilde{\pi}|p) \right] \geq \mathbb{E}_{p \sim P} \left[ L_\pi(\tilde{\pi}|p) \right] - \frac{2\lambda\gamma}{(1-\gamma)^2} \beta^2 \tag{43}$$

We now turn to optimize the RHS of (43) to maximize the objective in (6) under constraints $\alpha \leq \delta_1$:

$$\max_{\tilde{\pi}} \mathbb{E}_{p \sim P} \left[ L_\pi(\tilde{\pi}|p) \right] - \frac{2\lambda\gamma}{(1-\gamma)^2} \beta^2 \quad \text{s.t.} \quad \alpha \leq \delta_1. \tag{44}$$

Note that we have:

$$\alpha = \max_t \mathbb{E}_{s' \sim P^t(\cdot|p_w)} D_{TV}(\pi(a|s') \| \tilde{\pi}(a|s')) \leq \max_{s'} D_{TV}(\pi(a|s') \| \tilde{\pi}(a|s')) = \beta. \tag{45}$$

Following the approximation in Schulman et al. (2015), (44) can be equivalently transformed to:

$$\mathbb{E}_{p \sim P} \left[ \mathbb{E}_{s \sim P_\pi(\cdot|p), a \sim \pi(\cdot|s)} \left[ \frac{\tilde{\pi}(a|s)}{\pi(a|s)} A_\pi(s, a) \right] \right] \quad s.t. \quad \beta \leq \delta, \tag{46}$$

which can be solved using PPO (Schulman et al., 2017).

## A.4 Proof of Theorem 2

*Proof.* Denote $H(\pi^k \| \pi^{k+1}) \triangleq \max_t \mathbb{E}_{s' \sim P^t(\cdot | p_w)} D_{TV}(\pi^k(a|s') \| \pi^{k+1}(a|s'))$. Updating $\pi_k$ to $\pi_{k+1}$ at each iteration $k$ and following Theorem 1, we have

$$\eta(\pi_{k+1}|p_w^k) \geq \mathbb{E}_{p \sim P^{k+1}} \left[ \eta(\pi_{k+1}|p) - \frac{2|r|_{\max}\gamma\epsilon(p\|p_w^k)}{(1-\gamma)^2} \right] - \frac{4|r|_{max}H(\pi_k\|\pi_{k+1})}{(1-\gamma)^2}. \qquad (47)$$

Since $P^{k+1}$ and $\pi_{k+1}$ are obtained by maximizing the RHS of (3), we have

$$\mathbb{E}_{p \sim P^{k+1}} \left[ \eta(\pi_{k+1}|p) - \frac{2|r|_{\max}\gamma\epsilon(p\|p_w^k)}{(1-\gamma)^2} \right] - \frac{4|r|_{max}H(\pi_k\|\pi_{k+1})}{(1-\gamma)^2}. \qquad (48)$$

$$\geq \mathbb{E}_{p \sim P^{k+1}} \left[ \eta(\pi_k|p) - \frac{2r_{max}\gamma\epsilon(p\|p_w^k)}{(1-\gamma)^2} \right] - \frac{4|r|_{\max}H(\pi_k\|\pi_k)}{(1-\gamma)^2} \qquad (49)$$

$$= \mathbb{E}_{p \sim P^{k+1}} \left[ \eta(\pi_k|p) - \frac{2r_{max}\gamma\epsilon(p\|p_w^k)}{(1-\gamma)^2} \right] \qquad (50)$$

From Line 9 in Algorithm 1, the environment selected for training satisfies:

$$\eta(\pi_k|p) - \frac{2|r|_{\max}\gamma\epsilon(p\|p_w^k)}{(1-\gamma)^2} \geq \eta(\pi_k|p_w^k) - \frac{2|r|_{\max}\gamma\epsilon(p_w^k\|p_w^k)}{(1-\gamma)^2} = \eta(\pi_k|p_w^k). \qquad (51)$$

Therefore, combining (47)-(51), we have:

$$\eta(\pi_{k+1}|p_w^{k+1}) \approx \eta(\pi_{k+1}|p_w^k) \geq \mathbb{E}_{p \sim P^{k+1}}[\eta(\pi_k|p_w^k)] = \eta(\pi_k|p_w^k). \qquad (52)$$

where the approximation is made under the assumption that the expected returns of worst-case environment between two iterations are similar, which stems from the trust region constraint we impose on the update step between current and new policies, and can also be validated from experiments. □

## A.5 Empirical Verification of Assumption in Theorem 2

To verify the assumption made in Theorem 2, in Fig. 3, we study how the parameters of environments with poor performance scatter in the parameter space with different dimensions. Specifically, we plot the heatmap of return for the range of Hopper environments used for training, achieved by using MRPO to update the policy between two iterations. It can be validated that at the iteration $k = 300$, the poorly performing environments of the two policies before and after the MRPO update concentrate in the same region, i.e., the area of small frictions. The same result can be observed for the iteration $k = 350$.

For example, as shown in Figs. 3(a) and 3(b), at iteration $k = 300$, $p_w^{300} = (750, 0.5)$, the MC estimation of $\eta(\pi_{300}|p_w^{300})$ is 487.6 and that of $\eta(\pi_{301}|p_w^{300})$ is 532.0. At iteration $k = 301$, $p_w^{301} = (1027.8, 0.5)$ and the MC estimation of $\eta(\pi_{301}|p_w^{301})$ is 517.6. As shown in Figs. 3(c) and 3(d), at iteration $k = 350$, $p_w^{350} = (861.1, 0.5)$, the MC estimation of $\eta(\pi_{350}|p_w^{350})$ is 385.9 and that of $\eta(\pi_{351}|p_w^{350})$ is 422.2. At iteration $k = 351$, $p_w^{351} = (750, 0.5)$ and the MC estimation of $\eta(\pi_{351}|p_w^{351})$ is 394.0. In both cases, the empirical results can support the assumption that we made in (52), i.e., the expected returns of worst-case environment between two iterations are similar.

## A.6 Practical Implementation of MRPO

It can be seen from the expression of $\epsilon(p_w\|p)$ that it measures the transition dynamics distance between the worst-case environment $p_w$ and a specific environment $p$. In simulator, the environment's transition dynamics would vary with the environment parameters, such as the friction and mass in robot simulation. Hence the difference between environment parameters can reflect the distance between the transition dynamics. In addition, if we use in practice the penalty coefficient of $\epsilon(p_w\|p)$ as recommended by Theorem 1, the subset $\mathbb{T}$ would be very small. Therefore, we integrate it with $L_p$ in (9) as a tunable hyperparameter $\kappa$, and propose a practical version of MRPO in Algorithm 2.

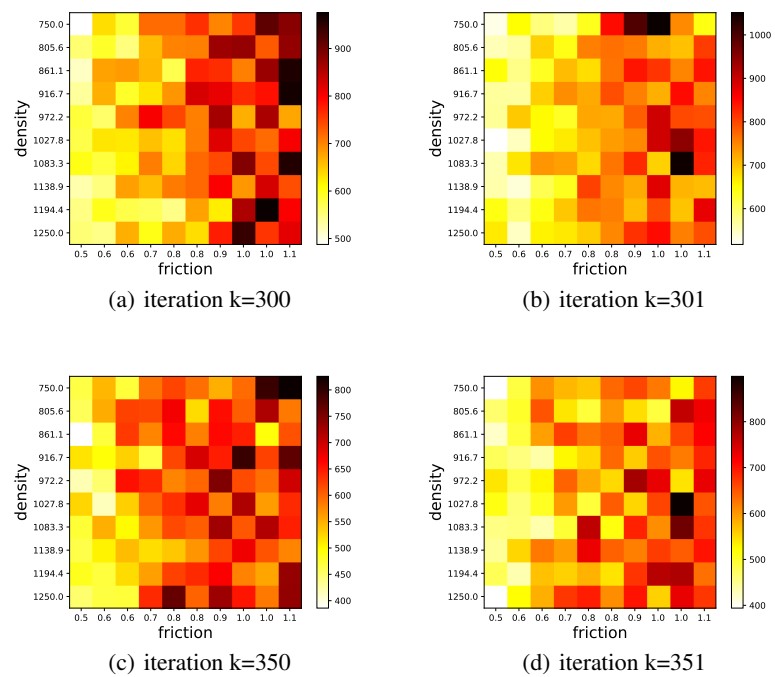

(a) iteration k=300

(b) iteration k=301

(c) iteration k=350

(d) iteration k=351

Figure 3: Heatmaps of return between policy update at iterations $k = 300$ and $k = 350$, using MRPO on Hopper.

---

**Algorithm 2** Practical Inplementation of Monotonic Robust Policy Optimization

---

1: Initialize policy $\pi_0$, uniform distribution of environment parameters $U$, number of environment parameters sampled per iteration $M$, maximum number of iterations N and maximum episode length $T$.
2: **for** $k = 0$ to $N - 1$ **do**
3:     Sample a set of environment parameters $\{p_i\}_{i=0}^{M-1}$ according to $U$.
4:     **for** $i = 0$ to $M - 1$ **do**
5:         Sample $L$ trajectories $\{\boldsymbol{\tau}_{i,j}\}_{j=0}^{L-1}$ in environment $p_i$ using $\pi_k$.
6:         Determine $p_w^k = \arg\min_{p_i \in \{p_i\}_{i=0}^{M-1}} \sum_{j=0}^{L-1} G(\boldsymbol{\tau}_{i,j}|p_i)/L$.
7:         Compute $\hat{\mathcal{E}}'(p_i, \pi_k) = \frac{\sum_{j=0}^{L-1} G(\boldsymbol{\tau}_{i,j}|p_i)}{L} - \kappa\|p_i - p_w^k\|$ for environment $p_i$.
8:     **end for**
9:     Select trajectory set $\mathbb{T} = \{\boldsymbol{\tau}_i : \hat{\mathcal{E}}'(p_i, \pi_k) \geq \hat{\mathcal{E}}'(p_w^k, \pi_k)\}$.
10:     Use PPO for policy optimization on $\mathbb{T}$ to get the updated policy $\pi_{k+1}$.
11: **end for**

---

### A.7 BOUNDED REWARD FUNCTION CONDITION IN ROBOT CONTROL TASKS

In Theorem 1, we state the condition that reward function is bounded. Referring to the source code of OpenAI gym (Brockman et al., 2016), the reward function for the five robot control tasks evaluated in this paper are listed below.

Hopper and Walker2d:

$$R = x_{t+1} - x_t + b - 0.001|a_t|^2;$$

Halfcheetah:

$$R = x_{t+1} - x_t - 0.001|a_t|^2;$$

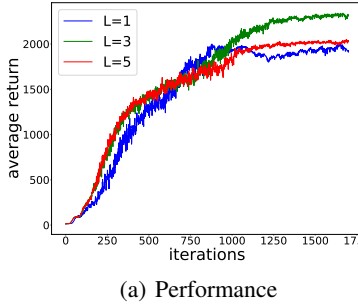 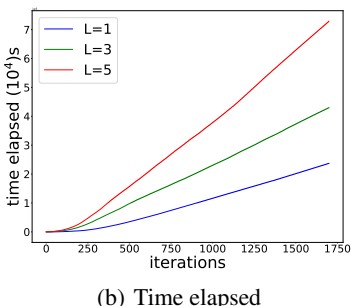

(a) Performance                    (b) Time elapsed

Figure 4:  (a) Training curves of average return of MRPO on Hopper with different $L$; (b) Time elapsed versus number of iterations curves during training.

Cartpole:

$$R = 1, \quad \text{if the pole does not fall down;}$$

InvertedDoublePendulum:

$$R = b - c_{dist} - c_{vel}.$$

In Hopper, Walker2d and Halfcheetah, $x_{t+1}$ and $x_t$ denote the positions of the robot at timestep $t+1$ and $t$, respectively. For Hopper and Walker2d, $b \in \{0, 1\}$, and $b$ equals $0$ when the robot falls down or $1$ otherwise. The squared norm of action represents the energy cost of the system. Since the maximum distance that the robot can move in one timestep and the energy cost by taking an action at each timestep are bounded, these three tasks all have the bounded reward function. In Cartpole, the reward is always $1$. In InvertedDoublePendulum, $b$ equals $0$ when the pendulum falls down or $10$ otherwise, $c_{dist}$ is the distance between the robot and the centre, and $c_{vel}$ is the weighted sum of the two pendulum's angular velocities. Since all the three parameters $b$, $c_{dist}$ and $c_{vel}$ are physically bounded, the reward function, as a linear combination of them, is also bounded.

A.8   ANALYSIS OF THE MONTE CARLO ESTIMATION OF $\eta(\pi|p)$

In Theorem 1, the worst-case environment parameter $p_w$ needs to be selected according to the expected cumulative discounted reward $\eta(\pi|p)$ of environment $p$. However, $\eta(\pi|p)$ is infeasible to get in the practical implementation. Therefore, as a commonly used alternative approach as in (Rajeswaran et al., 2017), we use the mean of the cumulative discounted reward of $L$ sampled trajectories $\sum_{j=0}^{L-1} G(\boldsymbol{\tau}_{i,j}|p_i)/L$ to approximate the expectation $\eta(\pi|p_i) = \mathbb{E}_{\boldsymbol{\tau}}[G(\boldsymbol{\tau}|p_i)]$ of any environment $p_i$, by using Monte Carlo method. We then determined the worst-case environment $p_w$ based on $\sum_{j=0}^{L-1} G(\boldsymbol{\tau}_{i,j}|p_i)/L$ of a given set of environments $p_i{}_{i=0}^{M-1}$. In the following, we will analyze the impact of $L$ on the estimation error.

**Theoretical analysis of the impact of $L$:** Referring to Chebyshev's inequality, for any environment $p_i$ and any $\varepsilon \geq 0$, with probability of at least $1 - \frac{\sigma^2}{L\varepsilon^2}$, we have

$$\left| \frac{\sum_{j=0}^{L-1} G(\boldsymbol{\tau}_{i,j}|p_i)}{L} - \frac{\sum_{j=0}^{L-1} \mathbb{E}_{\boldsymbol{\tau}_{i,j}}[G(\boldsymbol{\tau}_{i,j}|p_i)]}{L} \right| = \left| \frac{\sum_{j=0}^{L-1} G(\boldsymbol{\tau}_{i,j}|p_i)}{L} - \eta(\pi|p_i) \right| \leq \varepsilon, \quad (53)$$

where $\sigma = Var(G(\boldsymbol{\tau})|p_i)$ is the variance of trajectory $\boldsymbol{\tau}$'s return. From the above equation, we find out that the variance of the return does affect the MC estimation of $\eta(\pi|p)$ and a larger $L$ can guarantee a higher probability for the convergence of $\sum_{j=0}^{L-1} G(\boldsymbol{\tau}_{i,j}|p_i)/L$ to $\eta(\pi|p_i)$.

**Empirical evaluation of the impact of $L$:** In practice, we conduct experiment of MRPO on Hopper with different choices of $L$. We find out that the a larger $L$ would not greatly affect the performance in terms of average return as shown in Fig. 4(a), but will significantly increase the training

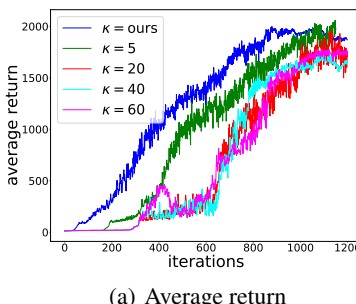
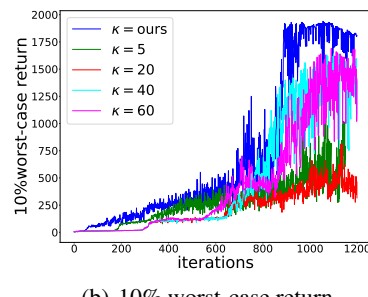

(a) Average return
(b) 10% worst-case return

Figure 5:   (a) Training curves of (a) average return and (b) 10% worst-case return of MRPO on Hopper with different $\kappa$.

time as shown in Fig. 4(b). In other words, for the same number of training iterations, a larger $L$ would consume significantly longer running time than a smaller $L$, while the performance is similar. Therefore, we set $L = 1$ in our practical implementation of MRPO to strike a trade-off between the approximation accuracy and time complexity in training.

### A.9   ANALYSIS OF THE LIPSCHITZ ASSUMPTION

In robot control tasks, classical optimal control methods commonly utilize the differential equation to formulate the dynamic model, which then indicates that the transition dynamics model is $L_p$-Lipschitz and this formulated dynamic function can be used to estimate the Lipschitz constant $L_p$.

For example, the inverted double pendulum, one of our test environments, can be viewed as a two-link pendulum system (Chang et al., 2019). To simplify the analysis, we illustrate here a single inverted pendulum, which is the basic unit that forms the inverted double pendulum system. The single inverted pendulum has two state variables $\theta$ and $\dot{\theta}$, and one control input $u$, where $\theta$ and $\dot{\theta}$ represent the angular position from the inverted position and the angular velocity, respectively, and $u$ is the torque. The system dynamics can therefore be described as

$$\ddot{\theta} = \frac{mgl\sin\theta + u - 0.1\dot{\theta}}{ml^2}, \tag{54}$$

where $m$ is the mass, $g$ is the Gravitational acceleration, and $l$ is the length of pendulum. In our setting, we may choose $m$ as the variable environment parameter $p$. Since the above system dynamics are differentiable w.r.t. $m$, it can be verified that the maximum value of the first derivative of the system dynamic model can be chosen as the Lipschitz constant $L_p$.

### A.10   HYPERPARAMETER $\kappa$

In Algorithm 2, when we update the sampling distribution $P$ for policy optimization, $\kappa$ is a hyperparameter that controls the trade-off between the expected cumulative discounted reward $\eta(\pi_k|p_i)$ and distance $\|p_i - p_w^k\|$ to the worst-case environment. Theoretically, a larger $\kappa$ means that the policy cares more about the poorly-performing environments, while a smaller $\kappa$ would par more attention to the average performance. As empirical evaluation, we conduct experiment of MRPO on Hopper with different choices of hyperparameter $\kappa$. The training curves of both average return and the 10% worst-case return are shown in Figs. 5(a) and 5(b), respectively. It can be verified that for the fixed value choice of $\kappa$, the curve of $\kappa = 5$ outperforms the curves of $\kappa = 20, 40, 60$ in terms of the average return in Fig. 5(a), while the curve of $\kappa = 60$ outperforms the curves of $\kappa = 5, 20, 40$ in terms of the 10% worst-case return in Fig. 5(b). In practical implementation, we gradually increase $\kappa$ to

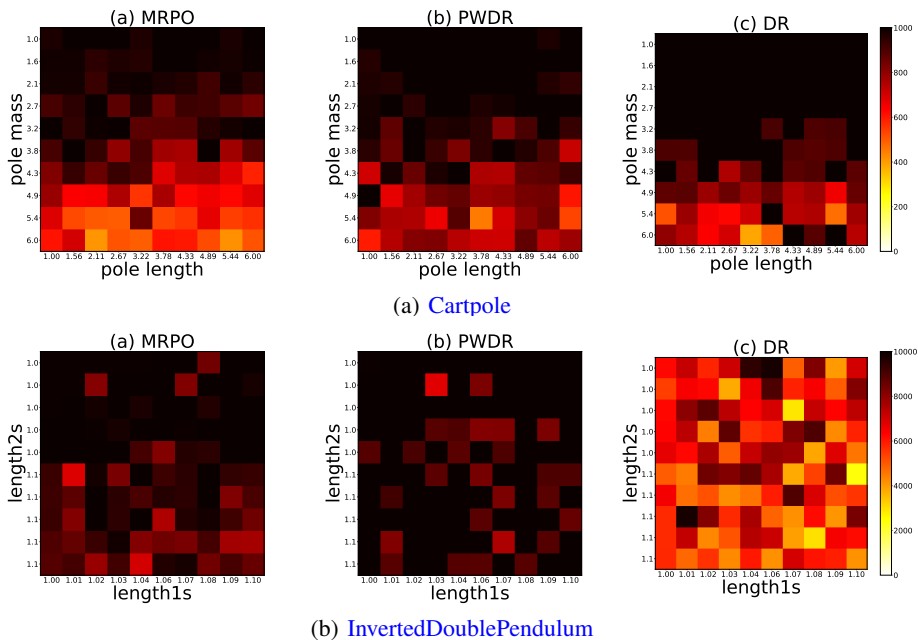

(a) Cartpole

(b) InvertedDoublePendulum

Figure 6: Heatmap of return in unseen environments on Cartpole and InvertedDoublePendulum with policies trained by MRPO, PW-DR and DR in the training environments.

a fixed high value. It can therefore strike a tradeoff between the average return and 10% worst-case return, demonstrating the best performance both in Figs. 5(a) and 5(b).

## A.11 GENERALIZATION TO UNSEEN ENVIRONMENTS OF CARTPOLE AND INVERTEDDOUBLEPENDULUM

In Fig. 6, we show the comparison results of MPRO, PR-DR and DR on unseen environments for the other two benchmarks, Cartpole and InvertedDoublePendulum, to provide empirical support for the generalization capability of MRPO.

