# OpenReview forum: "Monotonic Robust Policy Optimization with Model Discrepancy"
_ICLR.cc/2021/Conference — Reject_

### Official Review · AnonReviewer4 · 2020-10-28
**Model discrepancy between enviornments plays a role in generalization**

**Rating:** 7
**Confidence:** 4

**Review:**

This paper focuses on the generalization issue in reinforcemetn leanring, specifically aims to address the problems of domain randomization(DR) technique. Different from standard DR which treats all the sample environment as equal, this paper proposed to improve the performance over all possible environments and the worst-case environment concurrently. This paper theoretically derives a lower bound for the worst-case performance of a given policy over all environment, and in practical, the proposed method, monotonic robust policy optimization(MRPO) carries out a two-step optimization to imporve the lower bound such as to maximize the averaged and worst-case policy perfomance.


This paper is well written and the key concept is clearly introduced. The Theorem.1 makes the connections between the averaged and the worst-case performance, such that maximizing the worst-case performance can be solved by maximizing the averaged performance problem with some trajectories from environments with both poor and good-enough performance. The emprical results also support the theorical analysis.

1. For Lemma 1: The conclusion is based on the assumption that the the worst case $\rho(\pi|p_w) - \max_p \rho(\pi|\rho)$ is bounded (Proof A.1). However, such equation does not strictly holds without bounded reward function. The author should stated the condition.

2. About the monotonic worst-case performance improvement theorem, the proof says "... the approximation is made under the assumption that the worst-case environment between two iterations are similar, which stems from the trust region constraint we impose on the update step between current and new policies...", however, the trust region constraint can only limit the difference between policy updates, the similarity between worst-case environments can not be promised.

3. In theorem 2, the fomula (50) and (51) in the proof, is this approximation reasonable? Since the policy is updated, the worst-case environment may have changed a lot. Similarly, if the updated policy changes very little, can we make $\pi_{new}=\pi_{old}$ ?

4. The experiments are slightly inadequate, the effects of tunable hyperparameter k should be further analyzed; In unseen environment, the MRPO algorithm is only tested on one environment.

---

> ### Author Response · Authors · 2020-11-25
> **Author Response to Bounded Reward Function Condition**
>
> First of all, would like to thank the reviewer for providing the detailed comments. Please see below our detailed responses to these comments, and corresponding revisions in the rebuttal version of our paper.
>
> $\\\\$
>
>
> Comment 1: "For Lemma 1: The conclusion is based on the assumption that the worst case $\rho(\pi|p_w) - \max_p \rho(\pi|\rho)$ is bounded (Proof A.1). However, such equation does not strictly holds without bounded reward function. The author should stated the condition."
>
> Response:
>
> 1) We would like to thank the reviewer for pointing out bounded reward function condition. In Theorem 1 of the rebuttal version, we have stated this bounded reward function condition.
>
> 2) In Appendix A.7 of the rebuttal version, we have also listed the reward functions of the five robot control tasks evaluated in this paper to support this condition, as follows.
>
> Referring to the source code of OpenAI gym, the reward function for the five robot control tasks evaluated in this paper are listed below.
>
> Hopper and Walker2d:
> \begin{align*}
>     R = x_{t+1} - x_t + b - 0.001\vert a_t \vert^2;
> \end{align*}
>
> Halfcheetah:
> \begin{align*}
>     R = x_{t+1} - x_t - 0.001\vert a_t \vert^2;
> \end{align*}
>
> Cartpole:
> \begin{align*}
>     R = 1, \quad \text{if the pole does not fall down};
> \end{align*}
>
> InvertedDoublePendulum:
> \begin{align*}
>     R = b - c_{dist} - c_{vel}.
> \end{align*}
> In Hopper, Walker2d and Halfcheetah, $x_{t+1}$ and $x_{t}$ denote the positions of the robot at timestep $t+1$ and $t$, respectively. For Hopper and Walker2d, $b\in \{0,1\}$, and $b$ equals $0$ when the robot falls down or $1$ otherwise. The squared norm of action represents the energy cost of the system. Since the maximum distance that the robot can move in one timestep and the energy cost by taking an action at each timestep are bounded, these three tasks all have the bounded reward function. In Cartpole, the reward is always $1$. In InvertedDoublePendulum, $b$ equals $0$ when the pendulum falls down or $10$ otherwise, $c_{dist}$ is the distance between the robot and the centre, and $c_{vel}$ is the weighted sum of the two pendulum's angular velocities. Since all the three parameters $b$, $c_{dist}$ and $c_{vel}$ are physically bounded, the reward function, as a linear combination of them, is also bounded.

---

> > ### Author Response · Authors · 2020-11-25
> > **Author Response to Assumption of Similar Worst-case Performance Between Two Iterations**
> >
> > Comment 2: "About the monotonic worst-case performance improvement theorem, the proof says '... the approximation is made under the assumption that the worst-case environment between two iterations are similar, which stems from the trust region constraint we impose on the update step between current and new policies ...', however, the trust region constraint can only limit the difference between policy updates, the similarity between worst-case environments can not be promised."
> >
> > "In theorem 2, the formula (50) and (51) in the proof, is this approximation reasonable? Since the policy is updated, the worst-case environment may have changed a lot. Similarly, if the updated policy changes very little, can we make $\pi_{new} = \pi_{old}$?"
> >
> > Response:
> >
> > We would like to thank the reviewer for pointing out this issue. In Appendix A.5, Fig. 3 of the rebuttal version, we have added new empirical evaluation of MRPO on Hopper to validate this assumption, as follows.
> >
> > To verify the assumption made in Theorem 2, in Fig. 3, we study how the parameters of environments with poor performance scatter in the parameter space with different dimensions. Specifically, we plot the heatmap of return for the range of Hopper environments used for training, achieved by using MRPO to update the policy between two iterations. It can be validated that at the iteration $k=300$, the poorly performing environments of the two policies before and after the MRPO update concentrate in the same region, i.e., the area of small frictions. The same result can be observed for the iteration $k=350$.
> >
> > For example, as shown in Figs. 3(a) and 3(b), at iteration $k=300$, $p_w^{300} = (750, 0.5)$, the MC estimation of $\eta(\pi_{300}\vert p_w^{300})$ is $487.6$ and that of $\eta(\pi_{301}\vert p_w^{300})$ is $532.0$. At iteration $k=301$, $p_w^{301} = (1027.8,0.5)$ and the MC estimation of $\eta(\pi_{301}\vert p_w^{301})$ is $517.6$. As shown in Figs. 3(c) and 3(d), at iteration $k=350$, $p_w^{350} = (861.1,0.5)$, the MC estimation of $\eta(\pi_{350}\vert p_w^{350})$ is $385.9$ and that  of $\eta(\pi_{351}\vert p_w^{350})$ is $422.2$. At iteration $k=351$, $p_w^{351} = (750,0.5)$ and the MC estimation of $\eta(\pi_{351}\vert p_w^{351})$ is $394.0$. In both cases, the empirical results can support the assumption that we made in Equation (52), i.e., the expected returns of worst-case environment between two iterations are similar.
> >
> > Please note that we have slightly modified the proof of Theorem 2 in Appendix A.4 to be consistent with the above empirical verification of assumption in Theorem 2.

---

> > > ### Author Response · Authors · 2020-11-25
> > > **Author Response to Concern on Experiments**
> > >
> > > Comment 3: "The experiments are slightly inadequate, the effects of tunable hyperparameter $\kappa$ should be further analyzed; In unseen environment, the MRPO algorithm is only tested on one environment."
> > >
> > > Response:
> > >
> > > We would like to thank the reviewer for this suggestion. In Section 4, Appendix A.10 and Appendix A.11 of the rebuttal version, we have added more empirical evaluations in the following three aspects.
> > >
> > > 1) Analysis of hyperparameter $\kappa$:  In Appendix A.10 of the rebuttal version, we have added the following theoretical analysis and empirical evaluation on the hyperparameter $\kappa$.
> > >
> > >     Theoretically, in Algorithm 2, $\kappa$ is a hyperparameter that controls the trade-off between the expected cumulative discounted reward $\eta(\pi_k|p_i) $ and distance $\Vert p_i - p^k_w \Vert$ to the worst-case environment. A larger $\kappa$ means that the policy cares more about the poorly-performing environments, while a smaller $\kappa$ would par more attention to the average performance. As empirical evaluation, we conduct experiment of MRPO on Hopper with different choices of hyperparameter $\kappa$. The training curves of both average return and the 10\% worst-case return are shown in Figs. 5(a) and 5(b) of the rebuttal version, respectively. It can be verified that for the fixed value choice of $\kappa$, the curve of $\kappa=5$ outperforms the curves of $\kappa=20, 40, 60$ in terms of the average return in Fig. 5(a), while the curve of $\kappa=60$ outperforms the curves of $\kappa=5, 20, 40$ in terms of the 10\% worst-case return in Fig. 5(b). In practical implementation, we gradually increase $\kappa$ to a fixed high value. It can therefore strike a tradeoff between the average return and 10\% worst-case return, demonstrating the best performance both in Figs. 5(a) and 5(b) of the rebuttal version.
> > >
> > > 2) Evaluations on other benchmarks: In Section 4 of the rebuttal version, we have evaluated MRPO on more mujoco benchmarks like InvertedDoublePendulum, and also on classical control task like Cartpole. In addition, we have enlarged the friction range from the original $[0.5, 1.1]$ to $[0.2, 2.5]$ to form a new setting, HalfcheetahBroadRange.
> > >
> > > 3) More evaluations on unseen environments: In Section 4 and Appendix A.11 of the rebuttal version, we have also shown the comparison results on unseen environments for other benchmarks (e.g., Walker2D, HalfCheetahBroadRange, InvertedDoublePendulum and Cartpole), to provide empirical support for the generalization capability of MRPO.

---

### Official Review · AnonReviewer3 · 2020-10-28
**An improvement of EPOpt based on a TRPO-like lower bound on worst-case cumulative policy reward**

**Rating:** 6
**Confidence:** 2

**Review:**

Motivated by the domain transfer problem in RL where policies are trained on simulators that may not reflect perfectly the reality, this paper propose a new policy optimization algorithm named MRPO that is expected to be robust to changes in the environment's dynamic.
The formal setting and the notations are the same as in EPOpt (Rajewara 2017): each variant of the environment is an MDP parametrized by a parameter p and the trained policy is expected to be robust to (adversarial) changes on p.
Instead of focusing on worst cases with a CEM-like procedure on p distribution like in EPOpt, the authors propose to divert the TRPO approximation bound into a safety bound.
Theorem 1 gives a TRPLO-like lower bound, Theorem 2 show that optimizing for the LHS of Theorem 1 inequality may not degrade the wort-case reward.
The experiments study both the 10% word-case returns and the avarge returns. They show that MRPO improves clearly from EPOpt (renamed PW-DR for the occasion), the improvement against simple uniform domain randomization is less significant.

Probably because this paper relies on notions gathered from both (Rajewara 2017) and (Schulman et al. 2017), I found the 8 pages of the main paper quite dense and hard to follow. The proofs in the appendix are however clearly detailed and easy to read. I checked integrally the proof of Theorem 1/3 without any difficulty.

This domain randomization model formally equivalent to a single (continuous) MDP where the the environment's dynamic is parametrized by the initial state distribution (for instance by enriching the MDP states by the p parameter).
It is therefore unclear to me that a specific algorithm is required for the specific case of parametrized MDPs.
What would be the performance of a generic CVaR algorithm like "Risk-constrained reinforcement learning with percentile risk criteria" (Chow et al. 2017) on this setting ?
I found the idea of diverting the TRPO approximation bound into a safety bound appealing. Applied to a single MDP it could lead to a CVaR variant of TRPO.

Minor remarks:
p3 detalis -< details
I found the \rho notation for cumulative reward a bit confusing especially when p is involved in the equations, maybe a \nu instead would improve readability ?
Experiments on non-free systems like Mujoco are not easily reproducible. A few experiments on free-to-use environments would improve the reproducibility of the paper.

---

> ### Author Response · Authors · 2020-11-25
> **Author Response to Concern on Application of Generic CVaR Algorithm**
>
> First of all, we would like to thank the reviewer for providing the detailed comments. Please see below our detailed responses to these comments, and corresponding revisions in the rebuttal version of our paper.
>
> $\\\\$
>
> Comment 1: "This domain randomization model formally equivalent to a single (continuous) MDP where the the environment's dynamic is parametrized by the initial state distribution (for instance by enriching the MDP states by the p parameter). It is therefore unclear to me that a specific algorithm is required for the specific case of parametrized MDPs. What would be the performance of a generic CVaR algorithm like "Risk-constrained reinforcement learning with percentile risk criteria" (Chow et al. 2017) on this setting? I found the idea of diverting the TRPO approximation bound into a safety bound appealing. Applied to a single MDP it could lead to a CVaR variant of TRPO."
>
> Response:
>
> Please note that as a representative of robust RL algorithms, EPOpt in (Rajeswaran et al., 2017) was in essence a generic CVaR algorithm in the parameterized MDP case, which aims to maximize the conditional value at risk, i.e., the expected reward over the subset of environments with the lowest expected reward.
>
> Specifically, the optimization problem that EPOpt aims to solve is as follows:
> \begin{align*}
>     \max_{\theta, y} \int_{\mathcal{F}(\theta)} \eta(\pi_{\theta}\vert p) P(p)dp \quad s.t.\quad Pr(\eta(\pi_{\theta} \vert p)\leq y) = \epsilon,
> \end{align*}
> where $\mathcal{F}(\theta) = \{ p\vert \eta(\pi_{\theta}\vert p) \leq y \}$ is the set of environment parameters that produce the worst $\epsilon$ percentile of expected returns, and $y$ is the $\epsilon$-quantile of expected return $\eta$. It can be seen that this optimization problem can be viewed as the CVaR optimization under the parametrized MDP. For practical implementation, EPOpt proposed to optimize the policy on the subset of trajectories from the worst $\epsilon$ percentile environments, which was essentially an approximation solution to CVaR problem under parametrized MDP.
>
> In this paper, the baseline PW-DR was the practical implementation of EPOpt algorithm. Through performance evaluation on five different robot control tasks, we could see that compared to PW-DR, the proposed MRPO improved both the average and worst-case performance in the training environment, and achieved a better generalization performance in the unseen environments.
>
> Reference: Aravind Rajeswaran, Sarvjeet Ghotra, Balaraman Ravindran, and Sergey Levine.  EPOpt: Learning robust neural network policies using model ensembles. 2017.

---

> > ### Author Response · Authors · 2020-11-25
> > **Author Response to Other Concerns on Typo, Notation, and Free-to-Use Environments**
> >
> > Comment 2: "Minor remarks: p3 detalis -$<$ details"
> >
> > Response:
> >
> > We would like to thank the reviewer for the careful proofreading, and have corrected this typo in the rebuttal version.
> >
> >
> > $\\\\$
> >
> >
> > Comment 3: "I found the $\rho$ notation for cumulative reward a bit confusing especially when $p$ is involved in the equations, maybe a $\nu$ instead would improve readability?"
> >
> > Response:
> >
> > We would like to thank the reviewer for the this suggestion, and have replaced $\rho$ with $\eta$ in the rebuttal version. Please note that we did not use the $\nu$ notation because it may be a bit confusing with the state value function $V_{\pi}(s)$, and also because $\eta$ is commonly used in the literature for the cumulative reward, such as in the EPOpt (Rajeswaran et al., 2017).
> >
> >
> > $\\\\$
> >
> > Comment 4: " Experiments on non-free systems like Mujoco are not easily reproducible. A few experiments on free-to-use environments would improve the reproducibility of the paper."
> >
> > Response:
> >
> > 1) Please note that the environments used in our experiments on were all implemented based on Roboschool, which is an open-source software and free-to-use. The link for accessing Roboschool is https://openai.com/blog/roboschool/.
> >
> > 2) According the the reviewer's suggestion, we have also evaluated the proposed MRPO algorithm in Cartpole, which is an open-source classical control task.

---

### Official Review · AnonReviewer1 · 2020-10-28
**interesting algorithm with theoretical support**

**Rating:** 5
**Confidence:** 4

**Review:**

summary:
This paper introduces Monotonic Robust Policy Optimization (MRPO), an RL algorithm that aims to jointly optimize policy and domain sampling distribution, with the goal of improving policy performance for both average and worst-case scenarios and addressing the model discrepancy between the training and target environments. They derive a lower bound for the worst-case performance, which comprises the average performance, policy change, and the statistical distance between the worst and average case environments. A TRPO-like monotonic performance improvement guarantee is provided for the worst-case expected return. Finally, a practical approximation to MRPO is proposed, which imposes the assumption on Lipschitz continuity with respect to the environment parameters and circumvents the estimation of total variation distance between the worst-case environment and the sampled environment. Experiments are conducted on three control tasks with diverse transition dynamics parameters, where MRPO could improve both average and worst-case performance in the training environments, and it shows better generalization to the unseen test environments than baseline algorithms.


pros:
- The theoretical analysis is provided, which shows the relationship between the worst-case and average performance for the first time.

- The algorithm is backed by the theoretical guarantee of monotonic worst-case performance improvement.


cons:
- The assumption that the transition dynamics model is L-Lipschitz with respect to the environment parameter seems to be strong.

- Some of the experimental results are not convincing. For example, in Figure 1f, MRPO underperforms DR, even if DR does not consider the worst-case performance during optimization at all.


comments and questions:
- How natural is the model's Lipschitz assumption? Are many real-world problems satisfying this assumption?

- In Figure 1, what does the shaded-area stand for? standard deviation? standard error? Also, it is not clear that MRPO outperforms other baselines statistically significantly.

- It seems that two dense layers are used to construct the policy and value networks in the experiments. Why was the recurrent (e.g. LSTM) policy not used? Since the recurrent policy can implicitly embed system identification, I think the performance of the DR baseline could have been improved with the use of the recurrent policy. It would be great to see the performance comparison when the recurrent policy is used for MRPO and baselines.

- For the experiments on generalization to unseen environments, only the results for Hopper is provided, which may not be sufficient to demonstrate the behavior of each algorithm. It would be great to provide the heatmap results for other domains, i.e. Walker and HalfCheetah.

- In Theorem 1, is $p_w$ is the worst-case parameter for $\pi$? or for $\tilde \pi$? It would be good if notation presents the dependence on the policy of $p_w$, e.g. $p_w^\pi$.

- In Algorithm 2, line 6: how can $p_w^k$ be found? (even before completing sampling the trajectories for each environment)

---

> ### Author Response · Authors · 2020-11-25
> **Author Response to Lipschitz Assumption**
>
> First of all, we would like to thank the reviewer for providing the detailed comments. Please see below our detailed responses to these comments, and corresponding revisions in the rebuttal version of our paper.
>
> $\\\\$
>
> Comment 1: "The assumption that the transition dynamics model is L-Lipschitz with respect to the environment parameter seems to be strong."
>
> "How natural is the model's Lipschitz assumption? Are many real-world problems satisfying this assumption?"
>
> Response:
>
> 1) Reason to make the Lipshitz assumption: In robot control tasks, classical optimal control methods commonly utilize the differential equation to formulate the dynamic model, which then indicates that the transition dynamics model is $L_p$-Lipschitz and this formulated dynamic function can be used to estimate the Lipschitz constant $L_p$.
>
>     For example, the inverted double pendulum, one of our newly added test environments, can be viewed as a two-link pendulum system (Chang et al.,2019). To simplify the analysis, we illustrate here a single inverted pendulum, which is the basic unit that forms the inverted double pendulum system. The single inverted pendulum has two state variables $\theta$ and $\dot{\theta}$, and one control input $u$, where $\theta$ and $\dot{\theta}$ represent the angular position from the inverted position and the angular velocity, respectively, and $u$ is the torque. The system dynamics can therefore be described as
> \begin{align}
>     \ddot{\theta} = \frac{mgl \sin{\theta} + u -0.1\dot{\theta}}{m l^2},
> \end{align}
> where $m$ is the mass, $g$ is the Gravitational acceleration, and $l$ is the length of pendulum. In our setting, we may choose $m$ as the variable environment parameter $p$. Since the above system dynamics are differentiable w.r.t. $m$, it can be verified that the maximum value of the first derivative of the system dynamic model can be chosen as the Lipschitz constant $L_p$.
>
> Reference: Chang, Ya-Chien, Nima Roohi, and Sicun Gao. "Neural Lyapunov control." Advances in Neural Information Processing Systems. 2019.
>
> 2) Relation between the Lipschitz constant and the hyperparameter $\kappa$: From (3), it can be seen that the second term of the bound provided in Theorem 1 is not only dependent on the expected distance $\epsilon(p_w \Vert p)$, but also on $\frac{2|r|_{\max}\gamma}{(1-\gamma)^2}$. Therefore, in the practical implementation (Algorithm 2, Line 7), the Lipschitz constant was integrated into the hyperparameter $\kappa$ which was the tunable hyperparameter during the experiment.
>
>     Theoretically, in Algorithm 2, $\kappa$ is a hyperparameter that controls the trade-off between the expected cumulative discounted reward $\eta(\pi_k|p_i) $ and distance $\Vert p_i - p^k_w \Vert$ to the worst-case environment. A larger $\kappa$ means that the policy cares more about the poorly-performing environments, while a smaller $\kappa$ would par more attention to the average performance. As empirical evaluation, we conduct experiment of MRPO on Hopper with different choices of hyperparameter $\kappa$. The training curves of both average return and the 10\% worst-case return are shown in Figs. 5(a) and 5(b) of the rebuttal version, respectively. It can be verified that for the fixed value choice of $\kappa$, the curve of $\kappa=5$ outperforms the curves of $\kappa=20, 40, 60$ in terms of the average return in Fig. 5(a), while the curve of $\kappa=60$ outperforms the curves of $\kappa=5, 20, 40$ in terms of the 10\% worst-case return in Fig. 5(b). In practical implementation, we gradually increase $\kappa$ to a fixed high value. It can therefore strike a tradeoff between the average return and 10\% worst-case return, demonstrating the best performance both in Figs. 5(a) and 5(b) of the rebuttal version.
>
> 3) Revision in the rebuttal version: We have added Appendix A.9 to analyze the Lipschtz assumption, and Appendix A.10 to study the hayperparameter $\kappa$.

---

> > ### Author Response · Authors · 2020-11-25
> > **Author Response to Concern on Experimental Results**
> >
> > Comment 2: "Some of the experimental results are not convincing. For example, in Figure 1f, MRPO underperforms DR, even if DR does not consider the worst-case performance during optimization at all."
> >
> > "In Figure 1, what does the shaded-area stand for? standard deviation? standard error? it is not clear that MRPO outperforms other baselines statistically significantly."
> >
> > Response:
> >
> > 1) Evaluation on HalfCheetah: In Fig. 1(f) of the original submission, it was seen that MRPO did not outperform DR. We hypothesized and thought that this was because the original parameter range we set for the Halfcheetah task (e.g., the friction range of $[0.5, 1.1]$) was too narrow to cause seriously poor performance on the $10\%$ worst-case environments. In the rebuttal version, we have validate this hypothesis and reported new experiment result of the Halfcheetah task by enlarging the friction range from $[0.5, 1.1]$ to $[0.2, 2.5]$, which was denoted as HalfcheetahBroadRange. The training curves of 10\% worst-case return have been shown in Fig. 1(l), which can demonstrate that MRPO outperforms the other baselines.
> >
> >
> > 2) Explanation on the shaded area in Figure 1: In training, we run each algorithm on all the environments for five different random seeds. In Figure 1, the solid curve was used to represent the average performance of each algorithm on all the five seeds, while the shaded-area denoted the standard error of the algorithms' performance on all the five seeds. In Section 4.1 of the rebuttal version, we have added corresponding clarification for the shaded-area.
> >
> > 3) More Evaluations on Cartpole and InvertedDoublePendulum: In Fig. 1 and Table 2 of the rebuttal version, we have also shown empirical evaluation for two new robot control tasks, Cartpole and InvertedDoublePendulum. The newly reported results could also validate that MRPO generally outperforms the other baselines in the Cartpole and InvertedDoublePendulum tasks.

---

> > > ### Author Response · Authors · 2020-11-25
> > > **Author Response to the Recurrent Policy**
> > >
> > > Comment 3: "It seems that two dense layers are used to construct the policy and value networks in the experiments. Why was the recurrent (e.g. LSTM) policy not used? Since the recurrent policy can implicitly embed system identification, I think the performance of the DR baseline could have been improved with the use of the recurrent policy. It would be great to see the performance comparison when the recurrent policy is used for MRPO and baselines."
> > >
> > > Response:
> > >
> > > Packer et al. (2018) have conducted extensive performance comparison when two network architectures were used for policy and value functions for many different baselines, including  PPO and EPOpt-PPO. i) The first network architecture was the feed-forward (FF) architecture of multi-layer perceptrons (MLP) with two hidden layers of 64 units each. ii) The second was the recurrent (RC) architecture. In the RC architecture, the policy and value functions were the outputs of two separate fully-connected layers on top of a one-hidden-layer RNN with LSTM cells of 256 units, and the RNN itself was on top of an MLP with two hidden layers of 256 units each. Their experiments (please refer to Table 2 in (Packer et al., 2018) for more detail) showed that for the same baseline, the utilization of the second RC architecture would significantly degrade the generalization performance in all the cases, as compared to using the first FF architecture. Therefore, in this paper, we adopted the first feed-forward network architecture with two hidden layers of 64 units each to construct the policy and value functions of MRPO and the baselines.
> > >
> > >
> > > Reference: Charles Packer, Katelyn Gao, Jernej Kos, Philipp Krahenbuhl, Vladlen Koltun, and Dawn Song. Assessing generalization in deep reinforcement learning. arXiv preprint arXiv:1810.12282, 2018.

---

> > > > ### Author Response · Authors · 2020-11-25
> > > > **Author Response to Unseen Environment Results for Other Tasks, Clarification in Theorem 1, and Selection of the Worst-case Parameter**
> > > >
> > > > Comment 4: "For the experiments on generalization to unseen environments, only the results for Hopper is provided, which may not be sufficient to demonstrate the behavior of each algorithm. It would be great to provide the heatmap results for other domains, i.e. Walker and HalfCheetah."
> > > >
> > > > Response:
> > > >
> > > > In Fig. 2 and Appendix A.11 of the rebuttal version, we have shown the comparison results on unseen environments for other tasks (e.g., Walker2D, HalfCheetahBroadRange, InvertedDoublePendulum and Cartpole), to provide empirical support for the generalization capability of MRPO.
> > > >
> > > >
> > > > $\\\\$
> > > >
> > > > Comment 5: "In Theorem 1, is $p_w$ is the worst-case parameter for $\pi$? or for $\tilde{\pi}$? It would be good if notation presents the dependence on the policy of $p_w$, e.g. $p_w^\pi$."
> > > >
> > > > Response:
> > > >
> > > > We would like to thank the reviewer for this suggestion and clarify that $p_w$ is the worst-case environment for $\pi$. In the rebuttal version, we have made this clarification in Theorem 1.
> > > >
> > > > $\\\\$
> > > >
> > > > Comment 6: "In Algorithm 2, line 6: how can $p_w$ be found? (even before completing sampling the trajectories for each environment)"
> > > >
> > > > Response:
> > > >
> > > > 1) Please note that we have modified Algorithm 2 according to Reviewer 2's comments, to clarify how to determine the worst-case environment $p_w$. In the modified version of Algorithm 2, we sampled $L$ trajectories for each environment in Line 5. Then, by using Monte Carlo estimation, we determined $p_w$ based on the mean of the cumulative discounted reward of these $L$ sampled trajectories (i.e., $\sum_{j=0}^{L-1}G(\tau_{i,j}\vert p_i)/L$) in Line 6.
> > > >
> > > > 2) In Theorem 1, the worst-case environment parameter $p_w$ needs to be selected according to the expected cumulative discounted reward $\eta(\pi_k \vert p)$ of each environment $p$, which is infeasible to get in the practical implementation. Therefore, as a commonly used alternative approach as in (Rajeswaran et al., 2017), we used Monte Carlo sampling of $\sum_{j=0}^{L-1}G(\tau_{i,j}\vert p_i)/L$ to estimate the expectation $\eta(\pi\vert p_i)=E_{\tau}\left[G(\tau\vert p_i) \right]$, where we samples $L$ trajectories $\{\tau_{i,j}\}_{j=0}^{L-1}$ . In Appendix A.8 of the rebuttal version, we have analyzed the Monte Carlo Estimation, and the impact of number of sampled trajectories $L$ both theoretically and empirically.
> > > >
> > > > Reference: Aravind Rajeswaran, Sarvjeet Ghotra, Balaraman Ravindran, and Sergey Levine.  EPOpt: Learning robust neural network policies using model ensembles. 2017.

---

### Official Review · AnonReviewer2 · 2020-11-04
**Review for Monotonic Robust Policy Optimization**

**Rating:** 4
**Confidence:** 3

**Review:**

In this paper, the authors proposed a more robust policy optimization method for domain randomization, by constraining the gap between the average performance of the whole range of environments and the performance of the worst-case environments. To achieve this, the author provide a lower bound for the worst-case performance, though the lower bound does not take the uncertainty of the finite samples into account.

In addition, the algorithm 1 proposed by authors requires to calculate a model discrepancy between $p_{w}$ and other environments $p_{i} \sim P$, which is impractical to estimate by samples if the discrepancy is total variation distance. To achieve this, the authors assumes that the transition is lipschitz, with the requirement of tunning lipschitz constant. For empirical evaluation, the author compare with PPO with DR and PW-DR on three continuous benchmark mujoco task, which demonstrate that MRPO has some advantage over the other two algorithms.

The followings are my detailed comments and questions:
- I feel that selecting the worst-case environment is one of the key challenging of the proposed algorithm. I did not find the description how to choose the $p_{w}$ given a set of environments $\{ p_{i} \}_{i=0}^{M-1}$. If the authors means that the expected return of the a single trajectory can be used to select the worst-case environment, then how do your algorithm can guarantee the expected return of the sampled trajectories is the exact performance of the environment? The author did not give finite sample high confidence upper bound for empirical mc estimation, and the selection of the worst case environment would be hard to implement in practical settings?

- How do you choose or estimate the lipschitz constant? If the lipschitz constant is not right, then the bound will not given any practical guidence here.

- It would be great if the authors can explain the gap between algorithm 2 and your practical implementation of using the 10% worst-case environments. If so, then the algorithm the authors use in the experiments can be viewed as directly select top performance trajectories to perform policy optimization, which I think the final algorithm is not consistent with your algorithm presented in the methodology part (please correct me if I am wrong about the final algorithm).

- The experiments do not give strong empirical support for the new algorithm. The authors only evaluate on three environments, which I think is not enough, can the authors add more mujoco benchmarks? Also from the current results, I can not conclude that MRPO is better than PPO-DR since the evaluated domain is only three. Further, can the authors run more iterations to make sure the algorithms converge? The curves now presented in the paper did not converge.


Overall I think there is a gap between the methodology presented in the paper and the final practical algorithm, and the  lower bound presented in the paper does not take the uncertainty caused by the finite samples into account, which will not give guidance to design empirical algorithms since the variance of the mc return of the policy is large.  Finally the evaluation of the algorithms have not been conducted thoroughly.

---

> ### Author Response · Authors · 2020-11-25
> **Author Response to Uncertainty Caused by the Finite Samples**
>
> First of all, we would like to thank the reviewer for providing the detailed comments. Please see below our detailed responses to these comments, and corresponding revisions in the rebuttal version of our paper.
>
> $\\\\$
>
> Comment 1:
> "the author provide a lower bound for the worst-case performance, ..., the lower bound presented in the paper does not take the uncertainty caused by the finite samples into account, which will not give guidance to design empirical algorithms since the variance of the mc return of the policy is large."
>
> "I feel that selecting the worst-case environment is one of the key challenging of the proposed algorithm. I did not find the description how to choose the $p_w$ given a set of environments ${p_i}^{M-1}_{i=0}$. If the authors means that the expected return of the a single trajectory can be used to select the worst-case environment, then how do your algorithm can guarantee the expected return of the sampled trajectories is the exact performance of the environment? The author did not give finite sample high confidence upper bound for empirical mc estimation, and the selection of the worst case environment would be hard to implement in practical settings? "
>
> Response:
>
> 1) Description on selection of $p_w$: In Theorem 1, the worst-case environment parameter $p_w$ needs to be selected according to the expected cumulative discounted reward $\eta(\pi\vert p)$ of environment $p$. Please note that in the rebuttal version, following Reviewer 3's suggestion, we have changed the notation from $\rho(\pi\vert p)$ in the original submission to $\eta(\pi\vert p)$ to denote this expected cumulative discounted reward, such that possible confusion with the environment parameter $p$ is avoided. However, $\eta(\pi\vert p)$ is infeasible to get in the practical implementation. Therefore, as a commonly used alternative approach as in (Rajeswaran et al., 2017), we used in Algorithms 1 and 2 the mean of the cumulative discounted reward of $L$ sampled trajectories $\sum_{j=0}^{L-1}G(\tau_{i,j}|p_i)/L$ to approximate the expectation $\eta(\pi| p_i)=E_{\tau}[G(\tau| p_i) ]$ of any environment $p_i$, by using Monte Carlo method. In the original submission, we followed the setting in (Rajeswaran et al., 2017) and let $L=1$, i.e., $G(\tau_{i,1}\vert p_i)$ of a single trajectory $\tau_{i,1}$ was used to estimate $\eta(\pi\vert p_i)$. We then determined the worst-case environment $p_w$ based on $G(\tau_{i,1}\vert p_i)$ of a given set of environments ${p_i}^{M-1}_{i=0}$. In the following, we will analyze the impact of $L$ on the estimation error.
>
> Reference: Aravind Rajeswaran, Sarvjeet Ghotra, Balaraman Ravindran, and Sergey Levine.  EPOpt: Learning robust neural network policies using model ensembles. 2017.
>
>
> 2) Theoretical analysis of the impact of $L$: Referring to Chebyshev's inequality, for any environment $p_i$ and any $\varepsilon \geq 0$, with probability of at least $1-\frac{\sigma^2}{L\varepsilon^2}$, we have
> $  \left\vert  \frac{\sum_{j=0}^{L-1}G(\tau_{i,j}\vert p_i)}{L} -\frac{\sum_{j=0}^{L-1}E_{\tau_{i,j}}[G(\tau_{i,j}\vert p_i)]}{L} \right\vert =  \left\vert  \frac{\sum_{j=0}^{L-1}G(\tau_{i,j}\vert p_i)}{L} -\eta(\pi\vert p_i)\right\vert \leq \varepsilon, $ where $\sigma=Var(G(\tau\vert p_i))$ is the variance of trajectory $\tau$'s return. From the above equation, we find out that the variance of the return does affect the MC estimation of $\eta(\pi\vert p)$ and a larger $L$ can guarantee a higher probability for the convergence of $\sum_{j=0}^{L-1}G(\tau_{i,j}\vert p_i)/L$ to $\eta(\pi\vert p_i)$.
>
>
> 3) Empirical evaluation of the impact of $L$: In practice, we have conducted experiment of MRPO on Hopper with different choices of $L$. We found out that the a larger $L$ would not greatly affect the performance in terms of average return as shown in Fig. 4(a) in the rebuttal version, but would significantly increase the training time as shown in Fig. 4(b) in the rebuttal version. In other words, for the same number of training iterations, a larger $L$ would consume significantly longer running time than a smaller $L$, while the performance is similar. Therefore, we set $L=1$ in our practical implementation of MRPO to strike a trade-off between the approximation accuracy and time complexity in training.
>
>
> 4) Revision in the rebuttal version: We have modified Algorithms 1 and 2 to clarify how to select the worst-case environment $p_w$. We have also added Appendix A.8 to analyze the Monte Carlo Estimation of $\eta(\pi\vert p)$, and the impact of number of sampled trajectories $L$ both theoretically and empirically.

---

> > ### Author Response · Authors · 2020-11-25
> > **Author Response to Lipschitz Assumption and Lipschitz Constant Tuning**
> >
> > Comment 2: "In addition, the algorithm 1 proposed by authors requires to calculate a model discrepancy between $p_w$ and other environments $p_i \sim P$, which is impractical to estimate by samples if the discrepancy is total variation distance. To achieve this, the authors assumes that the transition is lipschitz, with the requirement of tunning lipschitz constant."
> >
> > "How do you choose or estimate the lipschitz constant? If the lipschitz constant is not right, then the bound will not given any practical guidance here."
> >
> > Response:
> >
> > 1) Reason to make the Lipshitz assumption: In robot control tasks, classical optimal control methods commonly utilize the differential equation to formulate the dynamic model, which then indicates that the transition dynamics model is $L_p$-Lipschitz and this formulated dynamic function can be used to estimate the Lipschitz constant $L_p$.
> >
> >     For example, inverted double pendulum, one of our newly added test environments, can be viewed as a two-link pendulum system (Chang et al., Neural Lyapunov control, 2019). To simplify the analysis, we illustrate here a single inverted pendulum, which is the basic unit that forms the inverted double pendulum system. The single inverted pendulum has two state variables $\theta$ and $\dot{\theta}$, and one control input $u$, where $\theta$ and $\dot{\theta}$ represent the angular position from the inverted position and the angular velocity, respectively, and $u$ is the torque. The system dynamics can therefore be described as
> > \begin{align}
> >     \ddot{\theta} = \frac{mgl \sin{\theta} + u -0.1\dot{\theta}}{m l^2},
> > \end{align}
> > where $m$ is the mass, $g$ is the Gravitational acceleration, and $l$ is the length of pendulum. In our setting, we may choose $m$ as the variable environment parameter $p$. Since the above system dynamics are differentiable w.r.t. $m$, it can be verified that the maximum value of the first derivative of the system dynamic model can be chosen as the Lipschitz constant $L_p$.
> >
> > 2) Relation between the bound in Theorem 1 and the Lipshitz assumption: Guided from the bound proposed in Theorem 1, we formulated a constrained optimization problem in (4), where the second constraint constrained the expected distance over all the possible environments to the worst-case environment. Then, based on our theoretical derivation in the proof of Theorem 3 in Appendix A.2, we used TV distance between two environments to measure this expected distance, which was hard to estimate in practice. Alternatively, we were looking for a substitution variable that satisfied the following two properties: i) positively correlated to the TV distance and ii) easy-to-access. Since the environment dynamics were determined by the environment parameters which satisfied the Lipschitz continuity condition in many robot control tasks, we therefore utilized the distance between environment parameters to reflect the TV distance as in (8) and (9).
> >
> > 3) Relation between the Lipschitz constant and the hyperparameter $\kappa$: From (3), it can be seen that the second term of the bound provided in Theorem 1 is not only dependent on the expected distance $\epsilon(p_w \Vert p)$, but also on $\frac{2|r|_{\max}\gamma}{(1-\gamma)^2}$. Therefore, in the practical implementation (Algorithm 2, Line 7), the Lipschitz constant was integrated into the hyperparameter $\kappa$ which was the tunable hyperparameter during the experiment.
> >
> >     Theoretically, in Algorithm 2, $\kappa$ is a hyperparameter that controls the trade-off between the expected cumulative discounted reward $\eta(\pi_k|p_i) $ and distance $\Vert p_i - p^k_w \Vert$ to the worst-case environment. A larger $\kappa$ means that the policy cares more about the poorly-performing environments, while a smaller $\kappa$ would par more attention to the average performance. As empirical evaluation, we conduct experiment of MRPO on Hopper with different choices of hyperparameter $\kappa$. The training curves of both average return and the 10\% worst-case return are shown in Figs. 5(a) and 5(b) of the rebuttal version, respectively. It can be verified that for the fixed value choice of $\kappa$, the curve of $\kappa=5$ outperforms the curves of $\kappa=20, 40, 60$ in terms of the average return in Fig. 5(a), while the curve of $\kappa=60$ outperforms the curves of $\kappa=5, 20, 40$ in terms of the 10\% worst-case return in Fig. 5(b). In practical implementation, we gradually increase $\kappa$ to a fixed high value. It can therefore strike a tradeoff between average return and 10\% worst-case return, as shown in Figs. 5(a) and 5(b). Therefore, even if there is estimation error on $L_p$, it can be compensated in practice by tuning the hyperparameter $\kappa$.
> >
> > 4) Revision in the rebuttal version: We have added Appendix A.9 to analyze the Lipschtz assumption, and Appendix A.10 to study the hayperparameter $\kappa$.

---

> > > ### Author Response · Authors · 2020-11-25
> > > **Author Response to the Gap Between Algorithm 2 and Final Practical Implementation**
> > >
> > > Comment 3: "It would be great if the authors can explain the gap between algorithm 2 and your practical implementation of using the 10\% worst-case environments. If so, then the algorithm the authors use in the experiments can be viewed as directly select top performance trajectories to perform policy optimization, which I think the final algorithm is not consistent with your algorithm presented in the methodology part (please correct me if I am wrong about the final algorithm)."
> > >
> > > Response:
> > >
> > > 1) Reason for using the 10\% worst-case environments: In the robot control tasks that we tested in the experiments, the environment dynamics were determined by multiple factors, such as the density and friction. Under this circumstance, a policy may perform poorly in multiple different tuples of density and friction. In other words, a single worst-case environment usually may not represent all the environments where the current policy performs very poorly. Taking this into account, we therefore used the $10\%$ worst-case environments in the practical implementation to replace using of the single worst-case environment.
> > >
> > >
> > > 2) Trajectory selection criterion and consistency with the methodology: In the final practical implementation, we did not select trajectories by only referring to their performance. Instead, we selected the subset of trajectories for training by referring to Line 9 in Algorithm 2, where the use of single worst-case environment was replaced by using the $10\%$ worst-case environments to calculate $E' (p^k_w,\pi_k)$ for the aforementioned reason. From the expression of $E'(p_i,\pi_k)=\sum_{j=0}^{L-1}G(\tau_{i,j}\vert p_i)/L -\kappa\Vert p_i - p^k_w \Vert$, it can be seen that the trajectory selection is based on a trade-off between the performance and the distance to the worst-case environment, as we described in detail in the paragraph under Equation (5). Please also note that Algorithm 2 is consistent with Algorithm 1, with the only difference being applying the Lipschitz assumption. Therefore, we believed that the final practical algorithm was also consistent with Algorithm 1 in the methodology part.
> > >
> > >
> > > 3) Revision in the rebuttal version: In the beginning of Section 4, we have tried our best to clarify the reason why we used the 10\% worst-case environments instead of a single worst-case environment for practical implementation.

---

> > > > ### Author Response · Authors · 2020-11-25
> > > > **Author Response to Concern on Empirical Evaluation**
> > > >
> > > > Comment 4: "Finally the evaluation of the algorithms have not been conducted thoroughly."
> > > >
> > > > "The experiments do not give strong empirical support for the new algorithm. The authors only evaluate on three environments, which I think is not enough, can the authors add more mujoco benchmarks? Also from the current results, I can not conclude that MRPO is better than PPO-DR since the evaluated domain is only three. Further, can the authors run more iterations to make sure the algorithms converge? The curves now presented in the paper did not converge."
> > > >
> > > > Response:
> > > >
> > > > We would like to thank the reviewer for this suggestion. In Section 4 and Appendix A.11 of the rebuttal version, we have added more empirical evaluations in the following three aspects.
> > > >
> > > > 1) We have evaluated MRPO on more mujoco benchmarks like InvertedDoublePendulum, and also on classical control task like Cartpole. In addition, we have enlarged the friction range from the original $[0.5, 1.1]$ to $[0.2, 2.5]$ to form a new setting, denoted as HalfcheetahBroadRange.
> > > >
> > > > 2) In Fig. 1 of the rebuttal version, more iterations have been run to make sure that each algorithm converges.
> > > >
> > > > 3) We have also shown the comparison results on unseen environments for other benchmarks (e.g., Walker2D, HalfCheetahBroadRange, InvertedDoublePendulum and Cartpole), to provide empirical support for the generalization capability of MRPO.

---

### Author Response · Authors · 2020-11-25
**Overall Author Response to All the Anonymous Reviewers**

We would like to thank all the anonymous reviewers for their constructive comments to help us improve this paper. We have carefully considered all of them in the rebuttal version of our paper, where our changes are highlighted in blue.

Here, we would like to summarize the shared concerns from reviewers and the corresponding major revisions that we have made in the rebuttal version. For the detailed explanation, please also refer to our responses to each reviewer's comments.

$\\\\$

(C-1) Concerns on Empirical Evaluations

    R-1) Evaluation on HalfCheetah: In Fig. 1(f) of the original submission, it was seen that MRPO did not outperform DR. We hypothesized and thought that this was because the original parameter range we set for the Halfcheetah task (e.g., the friction range of $[0.5, 1.1]$) was too narrow to cause seriously poor performance on the $10\%$ worst-case environments. In the rebuttal version, we have validate this hypothesis and reported new experiment result of the Halfcheetah task by enlarging the friction range from $[0.5, 1.1]$ to $[0.2, 2.5]$, which was denoted as HalfcheetahBroadRange. The training curves of 10\% worst-case return have been shown in Fig. 1(l), which can demonstrate that MRPO outperforms the other baselines.

    R-2) More evaluation benchmarks: We have evaluated MRPO on more mujoco benchmarks like InvertedDoublePendulum, and also on classical control task like Cartpole. In addition, we have enlarged the friction range from the original $[0.5, 1.1]$ to $[0.2, 2.5]$ to form a new setting, denoted as HalfcheetahBroadRange.

    R-3) Evaluation on  unseen environments for other benchmarks: We have also shown the comparison results on unseen environments for other benchmarks (e.g., Walker2D, HalfCheetahBroadRange, InvertedDoublePendulum and Cartpole), to provide empirical support for the generalization capability of MRPO.

$\\\\$

(C-2) Concerns on Lipschitz Assumption and Hyperparameter $\kappa$'s Tuning

    R-4）We have added Appendix A.9 to analyze the Lipschtz assumption, and Appendix A.10 to study the hayperparameter $\kappa$.

$\\\\$

(C-3) Concerns on Monte Carlo Sampling and Estimation

    R-5) We have added Appendix A.8 to analyze the Monte Carlo Estimation of $\eta(\pi\vert p)$, and the impact of number of sampled trajectories $L$ both theoretically and empirically.

$\\\\$

(C4) Concerns on Selection of the Worst-case environment $p_w$

    R-6) We have modified Algorithms 1 and 2 to clarify how to select the worst-case environment $p_w$.

$\\\\$

(C5) Concerns on Bounded Reward Function Condition in Theorem 1

    R-7) In Theorem 1 of the rebuttal version, we have stated this bounded reward function condition. And in Appendix A.7, we have also listed the reward functions of the five robot control tasks evaluated in this paper to support this condition.

$\\\\$

(C6) Concerns on Assumption of Similar Worst-case Performance Between Two Iterations

    R-8) In Appendix A.5, we have added new empirical evaluation of MRPO on Hopper to validate this assumption.

---

### Decision · Program_Chairs · 2021-01-07
**Final Decision**

**Decision:**

Reject

**Comment:**

The paper tackles the problem of mitigating the effect of model discrepancies between the learning and deployment environments. In particular, the author focus on the worst-case possible performance. The paper has both an empirical and theoretical flavor. The algorithm they derived is backed by theoretical guarantees. There exists a gap between the theory presented and the final practical algorithm, which generated some elements of concern from the reviewers. Some of these issues (choice and sensitivity of the Lipschitz constant, in what cases can we make that assumption, choice of p_w, discrepancy between the theoretical proposal and the practical algorithm) are well addressed in the rebuttal. However, after careful examination of the reviews, the meta-reviewer is still not convinced that the paper meets the minimum requirements for acceptance, as many of the reviewers' initial concerns still remain.